# Learning from Missing Values: Encoding Missingness in Representation-Space for LSTM Time Series Forecasting

## Abstract

While many state-of-the-art techniques reconstruct incomplete time series datasets by replacing gaps with modeled estimates, we propose an alternative: encode missing values as an extremal sentinel value, allowing a prediction model to learn from the pattern of missingness. Incomplete data is a common problem in real-world time series forecasting, particularly in environmental monitoring where sensor failures can cause continuous gaps in data. This paper proposes the *Min-Std* method, a novel computationally efficient imputation strategy that encodes missingness in representation-space with an extremal statistical sentinel ($min - \sigma$) mapped to 0 under Min-Max scaling. The result is that instead of training a model on (possibly imprecise) estimates for missing data, we simply replace the missing value with a sentinel the model can recognize to mean 'uninformative'. By ensuring this sentinel is uniquely mapped to 0, the only 0 values the model will receive are missing values, creating an explicit and consistent representation-space missingness encoding. We compare prediction results using our Min-Std imputation strategy against 12 imputation methods (including Kalman Smoothing and MissForest) across 6 different transformations on 28 distinct environmental datasets. Friedman's nonparametric test and critical difference ranking demonstrate that Min-Std imputation consistently yields superior predicting performance (measured by KGE, NMSE, and F1 Score) compared to complex model-based alternatives while being orders of magnitude faster (e.g. 0.02s vs 500s+). Our findings suggest that single-channel, explicit representation-space encodings of missingness are preferable to reconstruction-based imputation.

## 1 Introduction

In real world applications, it is not uncommon to have a dataset with missing values. Communication failures, power outages, sensor degradation, and manual logging errors are all common causes of incompleteness (or missingness) from real-world sensor data (Khayati et al., 2020; Gan et al., 2023; Luo, 2022). Missing values can occur as isolated points (singletons) and/or as contiguous blocks (gaps) depending on the nature of the failure (Khayati et al., 2020; Khattab et al., 2023; Zhang et al., 2020). Environmental monitoring networks—hydrometric gauges, meteorological stations, and urban air-quality sensors— experience communication dropouts, power interruptions, calibration drift, and maintenance gaps that routinely create missing values, often as multi-hour to multi-week blocks rather than isolated points. These gaps can distort correlation, break temporal dependencies, and can bias downstream predictions if not handled appropriately.

Long Short-Term Memory (LSTM) networks are a popular architecture for multivariate time-series prediction due to their ability to capture long-range dependencies and nonlinear temporal relationships (Tian et al., 2018; Li, 2024). Because LSTMs expect finite inputs $x_t \in \mathbb{R}^d$, NaN imputation is a mandatory preprocessing step to maintain sequence continuity. NaN imputation (or data imputation) involves using a model and/or statistical properties of a dataset to create and insert values where data is missing; recreating the missing data as accurately as possible, or providing neutral placeholders to maintain sequence continuity. The chosen NaN imputation method significantly influences model performance, yet these influences remain poorly characterized across datasets and preprocessing pipelines. In addition to NaN imputation, LSTM data preprocessing requires data scaling (or transforming). However, transformation methods applied after NaN

imputation impact the imputed value - potentially changing sentinel 0s with other values. This interaction between imputation and transformation inspired the creation of a new NaN imputation method '*Min-Std*' presented in this paper (i.e. "Minimum minus Standard deviation."). Min-Std ensures imputed NaN values map to an extremal value whose extremity is preserved after monotonic transforms, maintaining its usefulness as a missingness flag. While standard deviation is often considered in tandem with variable mean, the mean minus std does not guarantee an extrema, while the minimum minus std does. The minimum, maximum, and standard deviation are calculated using training data, and those values are used to impute/transform the training, validation, and test set. Subtracting the standard deviation also helps ensure potential outliers in the validation or testing set are larger than this extremal value.

This novel Min-Std imputation method was motivated by the broader machine learning practice of using deliberately structured representations to identify uninformative or masked values. For example, the input dropout layer of an LSTM causes the model to set some activations to 0 to improve generalizability. While partially inspired by input dropout, the experiments in this paper do not use input dropout. Instead, the empirical claim tested here is: after Min-Max scaling, Min-Std creates a unique low-end representation-space sentinel that can function as a learnable missingness indicator. This simultaneously informs the model of whether a value is 'missing', similar to a binary missing mask that is often paired with input features. This allows the Min-Std imputation paired with MinMax scaler to function as a representation-space coding for "uninformative" inputs and simultaneously as a missingness mask. Min-Std will also ensure that the missing values are always 1) the lowest value (assuming monotonic transformations) and 2) separated from the distribution of real values by a statistical margin. These two properties allow the imputed value to work as a missing indicator even after transformations.

To test the influence of NaN imputation methods we selected environmental streamflow and air datasets containing "naturally" missing values, instead of artificially corrupting datasets to add missingness as found in Cini et al. (2022); Che et al. (2018). Artificial missingness frequently fails to reflect the patterns in missingness found in real data (Radosavljević et al., 2025) which has been shown to contain predictive information (Che et al., 2018). Real monitoring data arrive with natural gaps, and are thus an ideal candidate to analyze imputation-scaling interactions and their impact on time series prediction. Datasets include the *Grand River Watershed Streamflow (GRWS)* - a new collection of 17 paired hourly hydrometric and meteorological time series spanning 13 years Anonymous (2025), and the *Beijing Air Quality* dataset - an hourly record of six air pollutants and meteorological variables from 12 sites in Beijing, China spanning 4 years Zhang et al. (2017). This provides us with 29 separate time series datasets with natural missingness at various rates.

Model-based and generative imputers may introduce two related risks. First, a generative imputer may suffer from **mode collapse** (or modal collapse), producing overly typical values that underrepresent rare events or extremes. Second, training a downstream predictor on data with generated synthetic values creates a **model-collapse-like** risk. Min-Std avoids both reconstruction risks by encoding missingness directly as a sentinel rather than treating generated values as observations.

This paper benchmarks all combinations of thirteen NaN imputation methods and six transformation methods, tested on multiple datasets containing natural missingness. Each of the 78 combinations is evaluated through a consistent LSTM forecasting framework and analyzed using a composite performance indicator of three metrics: Kling–Gupta Efficiency (KGE), Normalized Mean Squared Error (NMSE), and a thresholded (99th percentile) F1 Score. Statistical significance is assessed using the Friedman test with Holm and Nemenyi post-hoc comparisons (Demšar, 2006). Through this design, we systematically evaluate how preprocessing decisions affect the downstream predictive skill of time series sequence LSTM models.

**The contributions of this work are as follows:**

1. We introduce a standardized benchmark for evaluating the joint effects of imputation and transformation on LSTM forecasting accuracy on environmental data under natural missingness.

2. We propose *Min–Std*, a sentinel-style imputation method that preserves missingness semantics through monotonic transformations.

3. We provide a new open-source dataset, **GRWS**, offering realistic, multivariate environmental time series with natural missingness for reproducible benchmarking.

4. We report statistically validated findings demonstrating the significance of preprocessing interactions for neural sequence models, challenging assumptions that scaling and imputation are independent.

## 2 Data Preprocessing

Before we can use any time-series data for training and evaluating a model it must first undergo some preprocessing steps. The prediction model requires data to be complete (i.e. fill any missing values), and transformed to a common scale/range. To test this, we apply a range of imputation strategies to fill missing values, and a variety of transformation methods to scale or normalize the data. These preprocessing steps are critical to ensure that the machine learning algorithm can make use of the data (Che et al., 2018; Luo, 2022). These preprocessing steps are applied to all variables (features/predictors and labels) within each of the train, validation, and test splits of every dataset.

Note that some imputation and transformation methods are data-driven procedures, and must be 'fit' before use. To fit we may need to determine the mean of a variable, calculate the standard deviation, determine weights of a KNN (K-Nearest Neighbors), etc. We fit the methods on the training set only, then apply the fitted method to each of the train, validation, and test sets. This ensures there is no data leakage from the testing or validation set into the training to maintain fair evaluation.

### 2.1 NaN Imputation Methods

We evaluate 13 imputation variants: 8 conventional baselines, 3 Min-Std buffer variants, and 2 hybrid Min-Std/model-based methods. These baseline methods span constant imputation (Zero, Mean), simple smoothing (Forward Fill, Rolling Mean), and complex model-based imputation (K-Nearest Neighbors, MICE, MissForest, and Kalman Smoothing). Detailed configurations and implementations for these baselines are provided in Appendix C. We implemented and evaluated several common and advanced strategies for imputing missing values, as well as one novel Constant Imputation strategy we refer to as *Min-Std* below. For specific details on each method see Appendix C. These methods are summarized as follows:

- **Constant Imputation**: These methods impute a constant value at every missing position, this may be called a "sentinel" or "flag" imputation. Some of these can be used as indicators that values are missing, and are not intended to attempt to find similar or "correct" values.

  - **Zero Imputation**
  - **Mean Imputation**

- **Smoothing**: These methods propagate existing values or smooth functions but are not predictive of the original missing value:

  - **Forward Fill (ffill) then Backward Fill**
  - **Rolling Mean Imputation**

- **Model-based Imputation**: These methods try to infer the most probable true value for each missing position. The fitting process of these data dependent methods is more substantial and may be quite computationally expensive (see Table 1).

  - **K-Nearest Neighbors (KNN)** (Khayati et al., 2020)
  - **Multiple Imputation by Chained Equations (MICE)** (Van Buuren & Groothuis-Oudshoorn, 2011)
  - **MissForest** (Stekhoven & Bühlmann, 2012)
  - **Kalman Smoothing/Filtering** (Kalman, 1960)

  KNN, MICE, and MissForest assign values in one variable by considering the values of multiple other variables simultaneously. This means they may infer structure between variables, while Kalman considers only the variable being imputed.

### 2.2 Min-Std Imputation

In this novel contribution, each missing value is encoded with a deliberately extreme sentinel value: the variable's **training minimum** value **minus one training standard deviation**. This technique ensures the imputed value is far outside (lower than) the range of observed training data. This is helpful for flagging missing values and can remain a useful NaN flag after performing a downstream transformation (assuming monotonic transformations).

Subtracting one standard deviation establishes a statistical 'buffer' between the missing values and the observed values; in this case we have $k = 1$ and set the sentinel value as:

$$x_{sentinel} = x_{\min} - (k \times \sigma)$$

where $x_{\min}$ and $\sigma$ are the minimum value and standard deviation, respectively, calculated solely on the training set. And $k$ is the buffer scaling factor.

When paired with Min-Max scaling, this sentinel value (being the global minimum) maps exactly to 0. This representational strategy preserves the identifiability of NaN locations and is reversible; the minimum values of each variable mark NaN locations. This is also a potential learnable indicator for the LSTM model; similar to the widely used binary missingness-mask often paired with data designed so a model may learn from missingness (Harutyunyan et al., 2019), and has been shown to improve prediction performance over standard imputation alone (Lipton et al., 2016). This improvement comes from the fact that 'natural' missingness is often not random, but contains informative patterns (Che et al., 2018) that a model may exploit, this is an often overlooked detail in papers using artificial missingness for testing.

Note that it is possible that a rare future value (validation or test set) could fall below the training minimum; the buffer provides a range of such values that will still be mapped to the range $[0, 1]$ by Min-Max. If the value was an extreme outlier, by more than a standard deviation of the train minimum, then it would be mapped to a value lower than the sentinel value (a buffer exceedance). Such extreme out-of-range anomalies would still be distinct from the sentinel value (unless they happen to lie at exactly $x_{\min} - (k \times \sigma)$), as they would be mapped to negative values. As $k$ controls the size of the buffer, larger values increase buffer size and may be used in cases where large non-stationarities are expected, while smaller $k$ values may be more appropriate for data with less variance. We consider also **Min-Std/4** and **Min-Std/2** where instead of subtracting a full standard deviation we set the scaling buffer $k = 0.25$ or $k = 0.5$ respectively. While we consider values $k \in \{0.25, 0.5, 1\}$, $k$ functions as a parameter and other values may be used for the specific dataset to ensure separation.

### 2.3 Combined Imputation Methods

As noted by others Chari et al. (2025); Durbin & Koopman (2012), during testing we noted that many of the Model-Based imputation methods mentioned above, such as Kalman Smoothing, degrade and lose accuracy with long contiguous gaps. This inspired us to develop a novel-hybrid approach in which we apply a double-pass approach to imputation: singleton NaNs are estimated using Kalman or MissForest, then contiguous blocks of more than one consecutive NaN are replaced by the Min-Std sentinel value. We refer to these as *Kalman&MinStd* and *MissForest&MinStd*. The minimum value and standard deviation are both calculated on the train set prior to fitting the paired model (Kalman or MissForest), then the model is fit on the training set. The hybrid approach attempts to preserve signal continuity for small gaps, while clearly encoding larger missing segments.

### 2.4 Data Transformation Methods / Scalers

After imputation, the data must be scaled or normalized to ensure that each variable contributes comparably to the model's learning process. This is particularly important for models that are sensitive to magnitude differences between input features, such as LSTMs and other neural architectures. We evaluate the following transformation methods implemented using sklearn 1.3.2 implementations Pedregosa et al. (2011):

- **Standard Scaling**

- **Min-Max Scaling**

- **Robust Scaling**

- **Power Transform** (Yeo & Johnson, 2000)

- **Quantile Transform**

- **Max-Abs Scaling**

## 3 Methods

### 3.1 LSTM Model

To measure how preprocessing impacts predictive performance under data with missingness, we need a model to make predictions. For this model we use a stacked multilayer LSTM built using Keras 2.13 and Tensorflow 2.13, and trained on Compute Canada Rorqual cluster.

#### 3.1.1 Model Architecture

We employed a three layer stacked long short-term memory (LSTM) network (Hochreiter & Schmidhuber, 1997) for sequence-to-sequence model. The input at each training instance is a sliding window of 24 time-steps with variable number of predictors/features (the number of features varies across datasets; see Section 4).

The network is comprised of three LSTM layers with 256 units with Glorot-uniform kernel initialization and orthogonal recurrent initialization. The following dropout and regularization values were selected via Bayesian optimization using Optuna (Akiba et al., 2019). Dropout (with rate $p = 0.2$) is applied after the first and second LSTM layers. A small recurrent regularization ($L_1 = 2.88 \times 10^{-6}$, $L_2 = 8.38 \times 10^{-6}$) was applied to mitigate overfitting, and additive Gaussian noise ($\sigma = 0.0089$) was injected at input to improve generalization. The final LSTM output is passed through a single dense layer followed by reshaping layer to represent the predicted target sequence. We use ReLU at the output as the targets are non-negative (flows, concentrations).

The model architecture is:
`Input` $(24, n_{\text{feat}}) \rightarrow \text{LSTM}(256) \rightarrow \text{Dropout}(0.2) \rightarrow \text{LSTM}(256) \rightarrow \text{Dropout}(0.2) \rightarrow \text{LSTM}(256) \rightarrow \text{Dense}(12) \rightarrow \text{Reshape}(12, 1)$.

#### 3.1.2 Training

Models were trained in Keras/Tensorflow using the mean absolute error (MAE) loss, Adam optimizer (Kingma, 2014), and batch size 256 (all selected by Optuna). Datasets were split 70% training, 15% validation, 15% testing. Early stopping (Prechelt, 2002) monitored validation loss with patience= 20 and a minimum of 20 epochs before monitoring, training for up to 100 epochs and restoring the best weights by validation loss.

Training ran on the *Rorqual* cluster (Compute Canada) with a $20GB$ MIG slice of an NVIDIA H100 ($80GB$ total), 5 CPU cores, and $20GB$ RAM; within that MIG slice we enabled NVIDIA Multi-Process Service (MPS) to train five models concurrently.

Every model is trained under identical conditions, varying only the preprocessing steps (NaN imputation and transformation) and the datasets. Each combination of preprocessing strategy and dataset was repeated 5 times, so for any given NaN imputation method, transformation method, and dataset, we train and test 5 models. We evaluate prediction accuracy (Section 3.2) for each forecasted step; that is, if the model predicts 12 time steps ahead, accuracy was computed separately for prediction at $t+1, t+2, t+3, \ldots, t+12$ separately and each stored. Note that while the model predicted 12 steps, all analysis in this paper are restricted to the first 4 time-steps ($t + 1, \ldots, t + 4$). Longer horizons showed marked skill degradation in some datasets, we restrict to a common short horizon to support fair, like-for-like comparison across preprocessing choices.

### 3.2  Model Performance Metrics

Trained models are used to predict labels for unseen data and assessed using performance metrics that reflect:

1. the magnitude of prediction errors relative to natural variability (*error magnitude*),

2. the ability to correctly detect extreme events (*extreme events*),

3. the overall similarity between observed and predicted values (*shape accuracy*).

Normalized Mean Squared Error (NMSE) (Chai & Draxler, 2014; Cao & Tay, 2003) quantifies the overall deviation from observed values, scaled by observed variance, to measure error magnitude (1). The F1 Score, evaluated at the 99th percentile threshold, assesses whether extreme events are predicted at the correct times, offering a temporal extreme-event-based metric (2). Finally, the Kling-Gupta Efficiency (KGE) (Gupta et al., 2009; Kling et al., 2012) is commonly used in streamflow prediction; it provides a composite assessment of correlation, bias, and variability, serving as a robust measure of shape accuracy (3). These three metrics offer a balanced, interpretable, and comprehensive framework to evaluate both continuous prediction quality and discrete event detection skill. More specific details of each metric and the composite score can be found in Appendix E.

To reduce result complexity and enable model comparison, we construct a single multi-criteria "Composite Indicator" (Greco et al., 2019; Nardo et al., 2005) which integrates the three core metrics: shape accuracy (KGE), magnitude accuracy (NMSE), and extreme event accuracy (F1 Score at the $99^{th}$ percentile). The composite score is computed as a weighted sum:

$$\text{Composite Score} = \frac{\max(\text{KGE}, 0) + (1 - \min(\text{NMSE}, 1)) + (\text{F1 Score}^{99})}{3} \tag{1}$$

Here, NMSE is transformed to skill-like form by subtracting from 1, ensuring that the Composite Score, and all components are positively oriented (higher is better). Metrics are clipped and normalized to the range $[0, 1]$; this remains partly compensatory, meaning deficits in one metric can be offset by gains in another. The clipping reflects a quality threshold: models with $KGE < 0$ or $NMSE > 1$ are worse than mean predictors, these extremely low values diminish interpretability and do not provide meaningful information when selecting between top performing models (since models worse than a mean predictor will not be considered among top performers).

Note that we cannot compare a predicted value to a NaN value, and we do not want to compare a predicted value to an imputed value for final analysis. So we do not evaluate the eventual testing set predictions where there was a missing label. During training, progress is measured with all training/validation data with the imputed values.

### 3.3  Critical Difference Ranking

Friedman's nonparametric test (Friedman, 1940; 1937) compares $k$ algorithms (or *treatments*) across $N$ datasets by ranking the algorithms within each dataset (block) and testing null hypothesis that the average ranks are equal (no winner). Within each block and algorithm, we aggregate the 5 repeated runs by the median of the composite score before ranking. When the test finds statistical difference we plot the algorithms' average ranks on a *Critical Difference* (CD) diagram (also called a Demšar Plot).

The average ranking alone is not enough to definitively assess the 'best' algorithms, as the average rank gaps may be too small to be confident that the order is not due to noise. For this reason Demšar Demšar (2006) suggests additional post-hoc tests; we apply Nemenyi all-pairs post-hoc test (Nemenyi, 1963) and Holm post-hoc test (Holm, 1979). The CD diagram (Demšar Plots) shown below plot each algorithm using the average ranks, with the lowest rank (best) on the left. The two algorithms differ significantly only if their average ranks differ by at least the CD (indicated by the CD bar in the plots). Otherwise they are not significantly different given the chosen significance level $\alpha = 0.1$, consistent with Demšar's examples (Demšar, 2006), and

research using Demšar recommendations and similar rank comparisons with $\alpha = 0.1$(Blanchard et al., 2010; Sechidis & Brown, 2018) as well as research using Nemenyi post-hoc test to compare stream flow prediction models (Lu et al., 2023). Although not employing an average ranking, these hydrological studies also treat $\alpha = 0.1$ as a commonly used significance level (Blöschl et al., 2019; Blum et al., 2019; Vidrio-Sahagún et al., 2025). We use a Holm step-down vs-best post-hoc test to indicate the 'co-best', comparing all $k-1$ methods against the top ranked algorithm. We treat horizons (different predicted time-steps) within a dataset as repeated measures; $N$ reflects correlated blocks.

## 4    Data Description

Two sets of time-series datasets were chosen to test these methods. Each is described below, for predictions we use an input sequence length of 24 time-steps (24 hours) $t-23, t-22, t-21, ..., t$, and predict the next 4 consecutive time-steps (hours) ahead $t+1, t+2, t+3, t+4$.

### 4.1    Grand River Watershed Streamflow (GRWS)

This set of 17 hydrometeorological datasets was built for streamflow or flood prediction by the first author Anonymous (2025). The datasets each contain time series of weather variables and river flow for locations within the Grand River watershed located in southwestern Ontario Canada. The full GRWS dataset archive contains 17 datasets. We utilized one dataset (Leggatt) exclusively for hyperparameter selection (train/validation portions only) and excluded it from our evaluations leaving 16 stations. This was to ensure we are testing on unseen distributions, and ensuring hyperparameter generalization.

The predicted value (label) represents the water discharge (or flow rate) of the river measured at a specific monitoring station on the Grand River watershed, each of the datasets is used to predict one of these flow rate labels. The discharge is measured in cubic meters per second ($m^3$/s) and represents the volume of water passing through a cross-section of the river per unit time. The inputs (features) represent the observed climate measurements (temperate, precipitation, and in some cases wind speed/direction) measured at the nearest monitoring station. This dataset contains hourly values over a 13-year period. A fairly recent flood, in June 2017, and within the time span of this dataset, saw flows of over $400\frac{m^3}{s}$ at the West Montrose station caused by a 125+mm rainfall over the Upper Grand River watershed; the usual summer flows at this location are closer to $4\frac{m^3}{s}$. The author, Anonymous (2025), details the watershed characteristics and details further in the dataset description documents. Appendix A also provides a reproduced summary of the dataset description, including variable descriptions (Table 2), and time series visualization Figure 4 of one dataset (referred to as Drayton).

The dataset was assembled by scraping and consolidating open data from the Grand River Conservation Authority (GRCA), and matching each streamflow station with its nearest weather station. This pairing provides a physically consistent view of watershed-scale dynamics and realistic temporal gaps arising from sensor outages and communication delays. The datasets and documentation have been released publicly to facilitate reproducibility and further research.

### 4.2    Beijing Air Quality

The second set of datasets used contains weather and air quality data in Beijing, and was first published by Zhang et al. (2017) and obtained from UCI (Chen, 2017). The predicted value (label) represents the measured concentration of PM2.5 in the air at a specific monitoring station. There are 12 stations located at separate sites in Beijing. The datasets contain hourly recordings from 01/03/2013-28/02/2017.

Appendix B also provides a summary of the dataset description. Table 5 details each of the variables in the datasets, which are consistent across all of the Air Quality datasets. Figure 5 shows an example of one dataset (referred to as Aotizhongxin), the figure contains a separate graph for each variable all aligned along a shared time axis, and this figure also indicates the Train/Validation/Test splits. Each variable with missing values has these indicated in its associated graph, with text giving some statistics and red markers to show the location of missing values in the time series.

### 4.3 Data Missingness

Each station from the two dataset categories above, and the associated time series of features contain missingness. Missingness rates varied from 0.00% to 2.79% (mean≈ 0.74%) in GRWS and 0.97% to 2.03% (mean≈ 1.47%) in Beijing Air Quality. Appendices A and B contain Tables 3 and 6 with a details breakdown of the missingness in the GRWS and Air Quality respectively.

Table 3 shows information about the GRWS gaps, indicating that missingness is overwhelmingly *gap-dominated* with ≈ 84–99% of missingness found in gaps ($\geq$ 2 consecutive NaNs) for most stations. Several stations (e.g. Beaverdale and Bridgeport) show strong structural gaps (multiple variables have simultaneous gaps), with a few having over 50% of the missingness occur in structural gaps. Longer gaps appear frequently in some datasets (e.g., Brantford has 72 gaps $\geq 24\,h$; Bridgeport-WQ-Station has 66), and the largest gaps reach many weeks to months (e.g., max $> 720h$). Conversely there are some stations with near-zero missingness, or near-zero structural missingness.

For the Beijing Air Quality set (Table 6), missingness is still gap dominated, but less so than GRWS, with ≈ 68-88% of missingness found in gaps. Air Quality appears to have fewer multi-variable gaps with the highest percentage of structural gaps being ≈ 60% as opposed to ≈ 98% highest in GRWS. Overall, these tables indicate the presence of *prolonged, contiguous gaps* and, at times, *multivariate concurrent gaps*, rather than mostly independent singletons (missing completely at random).

## 5 Results

Once models have been trained and tested using each (NaN imputation,transformation) pair, we narrow the field to find the pair that yields the best prediction results. We use a combination of box plots and Critical Difference (CD) diagrams of average ranks (Demšar plots) (Demšar, 2006). To start, we select the strongest transformation methods and drop the poorest performers to simplify the analysis.

In the Figures 2 and 3, any algorithm marked with a star is co-best (not significantly worse than the best rank under Holm vs-best). Any marked with a circle is significantly worse in rank than the best. Because Holm test is typically more powerful than Nemenyi, there are algorithms within 1 CD of the best that are not indicated as co-best, this is expected and we rely primarily on the Holm vs-best test, the CD bar in diagrams is used as descriptive rather than to select winners.

### 5.1 Transformation

Here we see our first glimpse of how preprocessing impacts prediction ability; we examine the Composite Scores of all algorithms grouped by transformation method. Figure 1, a set of box-and-whisker plots, shows the prediction scores of all tests grouped by transformation method, one plot per method. The median Composite Scores being: Min-Max 0.7369, Max-Abs 0.7272, Robust 0.5053, Power 0.4951, Quantile 0.4893, Standard 0.4000. These plots show the IQR (interquartile range, 25th-75th percentiles) of the Min-Max and Max-Abs transformation methods entirely above (better than) the IQR of all other transformation methods. And the maximum (best) result of Quantile, Robust, Power, and Standard are all within, or very close to, the IQR of the Min-Max and Max-Abs methods. Figure 2 shows a CD diagram where we treat the transformation method as the algorithm/treatment to rank the transformations alone. Figure 1 indicate the *maximum* accuracy achieved by the models using any of the Quantile, Robust, Power, and Standard transformation methods, at most match, and do not substantially exceed, the *typical* accuracy of the Min-Max or Max-Abs methods. And Figure 2 shows the Min-Max or Max-Abs ranked highest.

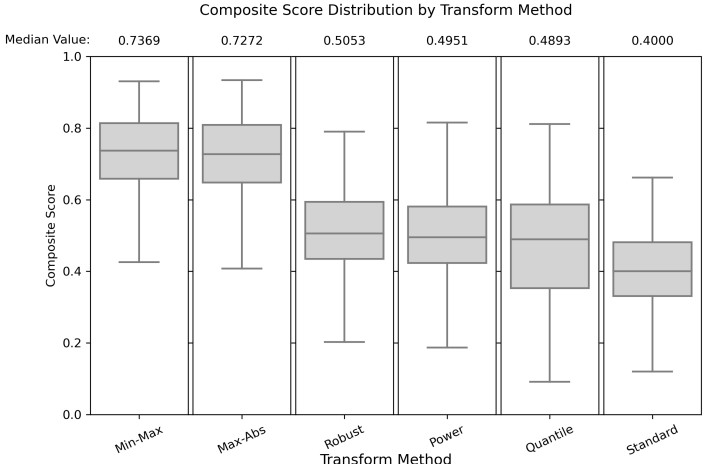

Figure 1: Each box represents the distribution of Composite Scores for the transformation method labeled below the plot. The boxes are ordered by median Composite Score, with the best median score on the left side. These median scores are displayed above the box plots.

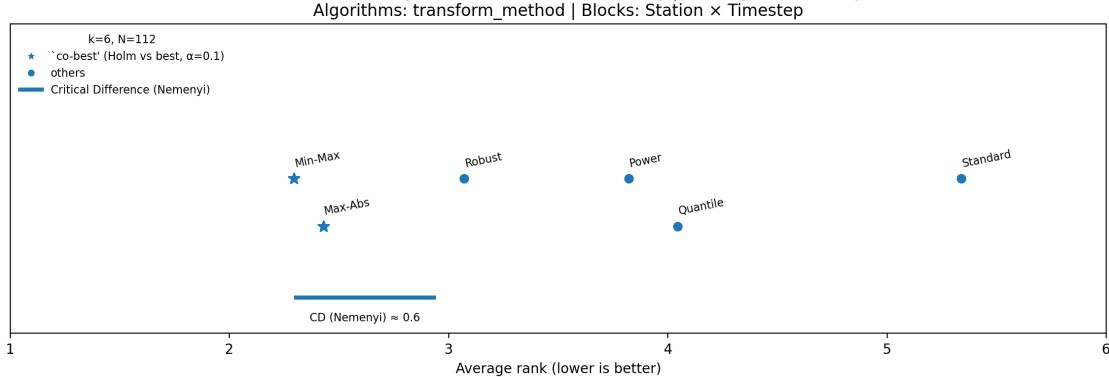

Figure 2: This Demšar/Critical Difference Plot shows the transformation method results ranked based on Composite Score. The blue horizontal bar shows the Critical Difference from the Nemenyi post-hoc test, in this case indicating the Min-Max and Max-Abs close enough to be tied for best. The stronger Holm post-hoc agrees and indicates the Min-Max and Max-Abs as 'co-best' transformation methods.

## 5.2 Imputation

Going forward we restrict analysis to these two transformations (Min-Max and Max-Abs) which will permit a much more decisive Critical Difference ranking. Demšar suggests a rule of thumb in Demšar (2006) that we need to have $k > 5$ algorithms and $N > 10$ datasets when applying Friedman tests. In Figure 3 we use the results from both GRWS ($N = 64$) and Air Quality ($N = 48$) for a total $N = 112$.

Figure 3: Average ranks over blocks (dataset x horizon) for all of both Air Quality and GRWS datasets combined. Friedman rejects equal-ranks ($\alpha = 0.10$). Blue bar: Nemenyi CD. Star = 'co-best' meaning not significantly worse than best under Holm vs-best. To improve readability, only the top-ranked combination is shown for each imputation method.

## 5.3 Computation Time

We also consider the computation time required for each method. We look at the average imputation times for Min-Std, Kalman, MissForest, KNN, MICE, Kalman&MinStd and MissForest&MinStd in Table 1.

Table 1: The average imputation times for various methods. This includes both the steps to fit the models on the training set, as well as the actual imputation process on the train, validation, and testing sets.

| Imputation Method | Mean Imputation Time (s) | |
|---|---|---|
| | Air Quality | GRWS |
| Min-Std | 0.02 | 0.03 |
| MICE | 3.02 | 4.05 |
| KNN | 4.79 | 5.22 |
| MissForest | 8.44 | 4.30 |
| MissForest&Min-Std | 8.40 | 4.31 |
| Kalman | 558.74 | 1213.78 |
| Kalman&Min-Std | 560.12 | 1217.24 |

## 6 Discussion

With large $k$ and smaller $N$, both Nemenyi and Holm lose power (the CD grows as $\sqrt{k(k+1)/(6N)}$) and struggle to identify a single best, meaning that most methods appear tied. To improve power we first looked at the results of all tests grouped by transformation methods, the box plot Figure 1 and the CD diagram Figure 2 both show the Min-Max and Max-Abs transformation methods significantly outperform the Power, Quantile, Robust, and Standard transformation methods, both indicating that Min-Max and Max-Abs result in significantly better prediction results, likely due to the non-Gaussian distribution inherent in environmental data. These results allows us to cull all other transformation methods reducing $k$ and allowing a more powerful analysis of the imputation/transformation combinations.

As we continue to comparing imputation/transformation combinations, in Figure 3 we see both the Nemenyi and Holm vs-best post-hoc tests identifying three methods as best: Min-Std, Kalman&MinStd and MissForest&MinStd. The two hybrid methods outperformed their model based components Kalman and MissForest, demonstrating that this combination is an improvement over the original model, but also the Min-Std is equally good as the combined, and better than the original model based methods.

To further contextualize these rankings, Appendix F contains additional plots underlying score distributions across the core performance metrics. These plots are separated by performance metric and show box plots for a selection of imputation/transformation combinations. Notably, the proposed Min-Std yields the tightest interquartile ranges and the fewest performance collapse outliers. This demonstrates that the representation-space encoding is not only highly accurate on average, but significantly more robust across diverse missingness patterns, dataset distributions, and random seeds than traditional imputation.

Table 1 looks at the computation time required to perform the imputations, we saw a considerable difference, with Min-Std taking a few hundredths of a second, and Kalman taking 10-20 minutes. In some cases, the time to complete the imputation process exceeded the time required to train the LSTM. So while prediction results after using Min-Std, Kalman&MinStd and MissForest&MinStd imputation were not significantly different, and all performed better than all other methods, we also consider that among these best three Min-Std is notably faster.

Reconstructing gaps using model-based or generative imputation risks distorting dataset probabilities and inducing model collapse (training a downstream model on synthetic data created by previous ML models) and modal collapse (under-representing rare modes or extremes). This may bias the downstream network toward artificial patterns. The proposed Min-Std method circumvents this danger by encoding missingness as an explicit sentinel rather than attempting to reconstruct with synthetic data. This ensures the model learns exclusively from true observed variance and the structural patterns of sensor failures. Furthermore, while specialized missingness-aware architectures (e.g., GRAPE, VAEAC) require massive computational overhead and network modifications, Min-Std provides an architecture-agnostic, near-zero-cost preprocessing alternative.

In Section 2.1 we discussed variations of the Min-Std imputer, where instead of subtracting one standard deviation ($k = 1$) from the minimum, we also tested $k = 0.25$ and $k = 0.5$ to use a sentinel value of $x_{\min} - (k \times \sigma)$. In our tests we found the full standard deviation performed better than fractional. We hypothesized that a full standard deviation ($k = 1$) could potentially cause loss of numerical resolution by compressing the range of real values after scaling. If this were the case, we would expect to see the quality of predictions reduce with larger $k$. With the datasets used in our testing, we see $k = 1$ consistently outperforming others, however, other values of $k$ may be proven better with different data ranges.

To examine whether this result is explained only by preventing real values from crossing the sentinel threshold, we performed a "buffer-exceedance" diagnostic (Appendix A Table 4 for GRWS, Appendix B Table 7 for Air Quality). For each tested $k$, we report the frequency of observed evaluation values falling below $x_{\min} - k\sigma$, the $k$ value required to eliminate such exceedances, and the transformation-space gap $g_k$ between the sentinel and the smallest observed training value. We also consider that values of $k > 1$ may be used to provide a larger 'buffer' if there are concerns of potential large non-stationarities in data. In addition to the train/validation/test split used in this study, we also performed a 5-fold inner cross-validation diagnostic on the training set to estimate buffer exceedances under resampling. The diagnostic shows that k=1 eliminates sentinel exceedances in the original validation/test holdout for both GRWS and Air Quality, and in an Air Quality inner-CV diagnostic. In GRWS inner-CV, a small number of exceedances remain at k=1, indicating that k=1 is not a mathematical guarantee under all resampling splits. However, k=1 substantially reduces exceedances relative to smaller k values, increases representation-space separation, and was empirically the best-performing tested value among $k \in \{0.25, 0.5, 1\}$. As noted in Section 2.1 $k$ is a parameter that may be set appropriate for the data.

## 7 Conclusions

This study identifies critical interactions between preprocessing strategies and prediction accuracy by benchmarking 78 imputation-transformation combinations across 28 distinct environmental datasets with natural missingness. Rather than comparing the accuracy of the imputations directly, we compared the quality of the resulting predictions to suggest a preprocessing strategy.

We publish the Grand River Watershed Streamflow time series datasets alongside this paper, a set of 17 ML-ready time series datasets. These datasets contain natural missingness ranging from $0.08\% - 2.79\%$,

each dataset contains 13 years of hourly recordings of streamflow data, paired with nearby weather data, and is intended for streamflow prediction tasks. We encourage researchers interested in environmental predictions, hydrology, or data imputation, to use the Grand River Watershed Streamflow datasets for time series predictions.

We also proposed a novel imputation method we call Min-Std, which replaces missing values with an extremal sentinel$= (x_{min} - \sigma)$. When paired with the Min-Max scaler, Min-Std creates a representation-space missingness encoding; imputed values are always 0, and no other values are 0. The choice to force the representation-space value of 0 was inspired by input-dropout commonly used with LSTM models, which randomly sets values to 0. The Min-Std strategy enforces a unified representation-space code, enabling the model to treat imputed missing values as uninformative, while simultaneously functioning as a missing indicator similar to a binary missing mask.

Among the scaling/transformation methods tested (Min-Max, Max-Abs, Quantile, Robust, Power, and Standard), Min-Max and Max-Abs both yielded prediction results far superior to the rest, while Standard scaling performed the worst by a significant margin. After subsequent tests, we observed that the best performing pairs of NaN imputation method and transformation method all used the Min-Max transformation method. When making predictions using an LSTM we recommend selecting between Min-Max and Max-Abs, with Min-Max being the preferred of the two.

Evaluation of imputation strategies (KNN, MICE, Kalman, MissForest, Forward Fill, Rolling Mean, Mean Fill, Zero Fill, and the novel method Min-Std) indicate that Min-Std yields the most accurate prediction results in both types of datasets. While hybrid methods (combining Kalman or MissForest with Min-Std) performed well, even outperforming their base Kalman/MissForest components, they were statistically indistinguishable from Min-Std alone. These hybrid methods also required significantly higher computation time than Min-Std.

We therefore recommend the Min-Std NaN imputation method paired with the Min-Max transformation method as a preprocessing strategy when making predictions with LSTM models on data that contains naturally occurring missingness. Evidence suggests that explicit representation-space encodings, such as Min-Std, offer a computationally and dimensionally efficient and high-performance alternative to traditional imputation methods that aim to reconstruct incomplete timeseries with model-based estimates, which prove difficult with gaps in stochastic data like precipitation. Our proposed strategy has a few benefits: in our testing it resulted in the best prediction results, it requires very little computation time, and it is extremely simple to identify which values had been missing after preprocessing.

We aim to continue this work by further analyzing the Min-Std method, but more generally by **consider Min-Std as a member of a broader class of Representation-Space Missingness Encoders (RSME)**: methods that replace missing values with deliberately chosen, non-reconstructive sentinel value whose purpose is to remain identifiable in the transformed feature space after preprocessing. Other members of this class might include upper bound sentinels such as max+std, sentinels less sensitive to rare spikes or outliers such as min-IQR or min-MAD, and target-space sentinels a desired transformed value is selected first, then the corresponding raw-space sentinel is solved for. We also consider causal gap-aware multi-sentinel RSME, where different sentinel values encode whether a missing value is the first missing value in a gap, or part of an ongoing missing run, with optional duration-dependent sentinels based on the time-since-last-seen value. We will also explore imputation methods in combination with multi-channel binary missing-masks and 'time-since-last-seen' features, with comparison of single-channel gap-aware RSME.

Future work will explore explainability techniques to characterize if/how the models learn to ignore the sentinel values, treat them as uninformative tokens, or if the models find informative predictive signal patterns in the missingness. We also plan to conduct additional LSTM experiments, including ablation studies with different dropout strategies, and to expand evaluation to Transformers and structured state-space models such as S4, which may respond differently to representation-space missingness encodings than recurrent architectures. Although this paper focuses on time-series forecasting using environmental data, future RSME work will examine non-environmental datasets as well as non-time-series applications including tabular prediction settings where missingness itself may carry predictive information.

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

## 8    Appendices

Appendix A and B contains detailed descriptions of the datasets used in this study, including variable descriptions (Tables 2 and 5), time series visualizations (Figures 4 and 5), and statistical details of the missingness in the datasets (Tables 3 and 6).

Tables 3 and 6 show various statistics regarding the missingness within each of the datasets. *Miss %* is the fraction of all cells that are NaN. *Gap %* is the share of NaN cells that occur inside gaps ($\geq 2$ consecutive NaNs) rather than as singletons. *Structural %* is the share of NaN cells where *two or more* variables are simultaneously inside such gaps (i.e., networked or concurrent outages). The Gap % and Structural % columns are computed over the total number of NaNs (e.g., Gap % = NaN cells in runs $\geq 2$ divided by all NaN cells. *Num Gaps* $\geq 24$ is the sum of long gaps (at least 24 consecutive hourly steps) aggregated per variable. *Gap (mean/med/max)* summarizes gap lengths (in hours) across all variables and gaps per station.

Figures 4 and 5 each contain: a subplot for each variable within one sample dataset (from GRWS (Drayton) and Beijing Air Quality dataset (Aotizhongxin) respectively) all aligned along a shared time axis, the Train/Validation/Test splits are indicated with dotted lines, and the distribution of NaNs in the variables indicated with red bars along the top of the subplots.

## A    GRWS: Extended Data Analysis

This appendix contains detailed descriptions of the GRWS datasets used in this study, including variable descriptions (Table 2), time series visualizations (Figure 4), and a table with statistical details of the missingness in the GRWS datasets (Table 3).

NOTE: The figures and tables in this Section are reproduced from the GRWS description Anonymous (2025) by the author as supplemental material. For further details and data access refer to that document. Note that not each of the datasets in GRWS contains the exact same variables.

Table 2: This table provides details for each of the variables in the GRWS dataset. Variables marked with ⋆ were not available for every location, and are not present in every dataset (for example Water Level is not contained in the 'Drayton' dataset visualization below). Each non-date variable has a code indicating the type of variable, QR for river flow, TA for air temperature, etc. These codes are also used in Figure 4 in the variable names above each subplot. The letter in the 'Subplot' column refers to which plot in Figure 4 (an example of one GRWS datasets called Drayton) represents this variable. The datasets all share three variables of air temperature (TA) statistics; these are the minimum, maximum, and mean air temperature across all TA time series for the watershed not just temperature for the nearest station.

| Variable | Details | Subplot |
|---|---|---|
| DateTime | Date/Time encoded with sine and cosine to represent cyclic patters: 'Time sin', 'Time cos', 'Day sin', 'Day cos', 'Year sin', 'Year cos'. | - |
| River Flow (QR) | The measured streamflow at the location in $\frac{m^3}{s}$. Used as the label/predicted value. | $A$ |
| Water Level (HG) ⋆ | The height above an arbitrary datum in $m$. | - |
| Min, Mean, Max Temperature (TA) | Wet bulb air temperature (in $°C$) statistics across *all weather* stations in the watershed | $B, C, D$ |
| Nearest Temperature (TA) | Wet bulb air temperature (in $°C$) measured at the *nearest* weather station. | $H$ |
| Nearest Precipitation (PN) | Tipping bucket precipitation measured in $mm$ at the nearest weather station. | $F$ |
| Nearest Wind Dir (UD) ⋆ | Wind direction in degrees at the nearest weather station. | $E$ |
| Nearest Wind Speed (US) ⋆ | Wind speed in km/h at the nearest weather station. | $G$ |

Table 3: A summary of the statistical properties of the missingness of each GRWS dataset. The most complete of these has only 5 values missing from 113,952, the dataset containing the most missing values is over 2.79% of values missing, the variable with the most missing values of the datasets is missing 11.97% of the values in that column. The shading simply indicates a high/med/low within the table, with dark grey cells being the largest 1/3 for the column, and white being lowest 1/3.

| Dataset GRWS | Miss % | Gap % | Structural % | Num Gaps $\geq 24$ | Gap (mean\|med\|max) |
|---|---|---|---|---|---|
| Beaverdale | 0.58% (3999) | 99.85% (3993) | 83.72% (3348) | 16 | 142.6\|45\|881 |
| Below-Elmira | 0.17% (1338) | 98.21% (1314) | 97.91% (1310) | 20 | 22.7\|12\|100 |
| Brantford | 1.49% (13608) | 98.97% (13468) | 4.35% (592) | 72 | 44.2\|11\|1464 |
| Bridgeport-WQ-Station | 2.79% (25437) | 99.38% (25279) | 52.78% (13426) | 66 | 122.1\|15\|6546 |
| Bridgeport | 2.52% (17230) | 99.09% (17074) | 97.35% (16773) | 14 | 258.7\|6\|7822 |
| Doon | 0.08% (531) | 90.40% (480) | 32.39% (172) | 8 | 17.1\|5\|107 |
| Drayton | 0.22% (1998) | 92.59% (1850) | 6.76% (135) | 15 | 14.3\|3\|217 |
| Dundalk | 0.00% (5) | 40.00% (2) | 0.00% (0) | 0 | 2.0\|2\|2 |
| Elmira | 0.12% (918) | 96.08% (882) | 95.75% (879) | 10 | 16.3\|8\|118 |
| Galt | 0.52% (4168) | 96.88% (4038) | 45.37% (1891) | 23 | 20.3\|7\|1424 |
| Glen-Allan | 0.25% (2534) | 93.61% (2372) | 55.29% (1401) | 19 | 19.9\|4\|364 |
| Hidden-Valley | 0.72% (4943) | 84.36% (4170) | 76.94% (3803) | 18 | 35.3\|2\|745 |
| Mount-Vernon | 0.64% (5129) | 97.76% (5014) | 1.44% (74) | 27 | 35.3\|7\|1444 |
| Salem | 1.34% (7625) | 97.14% (7407) | 0.47% (36) | 17 | 105.8\|4\|5634 |
| Victoria-Road | 0.18% (1419) | 97.11% (1378) | 22.27% (316) | 11 | 32.8\|4\|396 |
| West-Montrose | 0.17% (1380) | 96.67% (1334) | 96.38% (1330) | 12 | 51.3\|14\|259 |

Table 4: Min-Std buffer diagnostics for GRWS. "Below train min" counts observed evaluation values below $x_{\min}^{train}$. Sentinel exceedance counts observed evaluation values below $x_{\min}^{train} - k\sigma^{train}$. "$k_{\text{req,max}}$" is the train-standard-deviation buffer required to make sentinel exceedance zero across the analyzed variables for that split. Each $k$ cell reports total sentinel exceedance percentage, count, worst variable/fold exceedance percentage, and median Min-Max-space gap $g_k$. The Original Val+Test Holdout row combines the original validation and test splits for this diagnostic summary. Shading marks nonzero sentinel exceedances, with darker cells indicating larger total exceedance percentages.

| Split | Observed | Below train min | $k_{\text{req,max}}$ | $k = 0.25$ | $k = 0.50$ | $k = 1.00$ | $k = 2.00$ |
|---|---|---|---|---|---|---|---|
| Train 5-fold Inner CV | 7,765,939 | 0.4037% (31,348) | 2.625 | 0.0468% (3,636) max 15.293% $g$ 4.18% | 0.0210% (1,632) max 6.793% $g$ 8.02% | 0.0103% (803) max 6.040% $g$ 14.86% | 0.0063% (489) max 3.678% $g$ 25.87% |
| Original Val+Test Holdout | 4,009,037 | 0.0136% (547) | 0.321 | 0.0001% (5) max 0.023% $g$ 4.11% | 0.0000% (0) max 0.000% $g$ 7.90% | 0.0000% (0) max 0.000% $g$ 14.64% | 0.0000% (0) max 0.000% $g$ 25.54% |

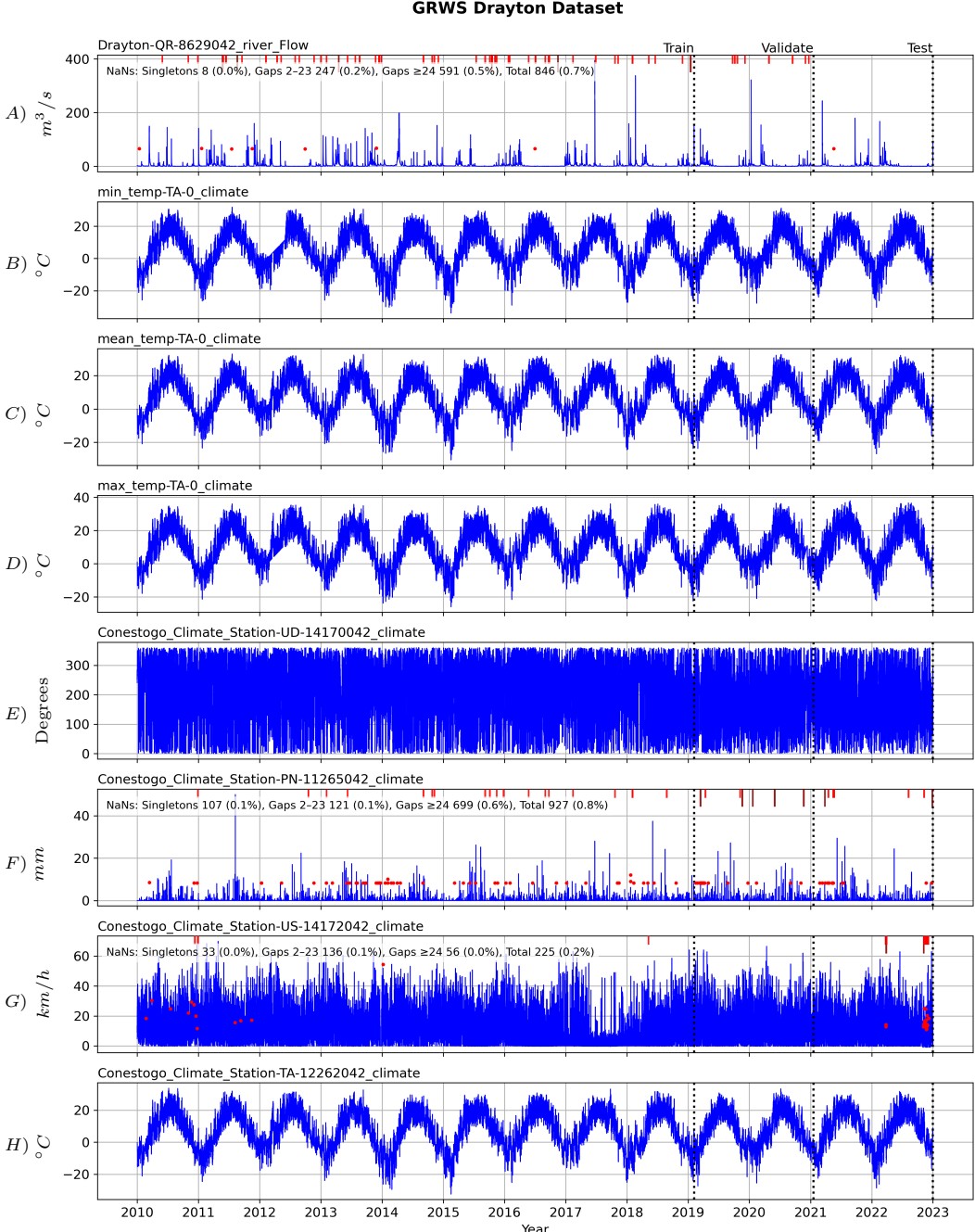

Figure 4: This plot shows the Drayton streamflow dataset, the topmost graph (*A*) with subplot named 'Drayton-QR-8629042_river_Flow' represents the values being predicted. Each subplot name follows a similar structure: the location of this monitoring station ('Drayton'), the variable code as described in Table 2 ('QR') meaning River Flow, the time series id assigned by the GRCA ('8629042'), 'river' means this is hydrological data vs climate, and finally 'Flow' is a tag used by the codebase to indicate that a variable a label. Red points on the graph indicate the presence of a singleton NaN, the red bars on the top indicate gaps in the data, with the larger darker bars being $\geq 24h$. The per-panel "NaNs:" line reports percentages relative to all timestamps in view (not to the NaN set). The plot also indicates the train/validation/test splits, with train start 2010-01-02, train end 2019-02-06 14:00:00, Validation end: 2021-01-18 19:00:00, Testing end: 2022-12-31 23:00:00. A total of 113952 time steps. The DateTime statistics are not included in this plot. This plot

# B   Beijing Air Quality: Extended Data Analysis

This appendix contains detailed descriptions of the Beijinig air quality datasets used in this study, including variable descriptions (Table 5), time series visualizations (Figure 5), and a table with statistical details of the missingness in the datasets (Table 6).

Table 5: This table provides details for each of the variables in the Air Quality dataset. The letter in the 'Subplot' column refers to which plot in Figure 5 (an example of one Air Quality datasets called Aotizhongxin) represents this variable.

| Variable | Details | Subplot |
|---|---|---|
| DateTime | Date/Time encoded with sine and cosine to represent cyclic patters: 'Time sin', 'Time cos', 'Day sin', 'Day cos', 'Year sin', 'Year cos'. | - |
| PM10 | Concentration of PM10 in $\mu g/m^3$. | A |
| SO2 | Sulfur Dioxide concentration in $\mu g/m^3$. | B |
| NO2 | Nitrogen Dioxide concentration in $\mu g/m^3$. | C |
| CO | Carbon Monoxide concentration in $\mu g/m^3$. | D |
| O3 | Ozone concentration in $\mu g/m^3$. | E |
| TEMP | Temperature in degrees Celsius. | F |
| PRES | Atmospheric pressure in hPa. | G |
| DEWP | Dew point temperature in degrees Celsius. | H |
| RAIN | Precipitation in mm. Almost always 0 in this dataset. | I |
| wdd | Wind direction, originally "NW", "ENE", etc. but has been converted to degrees 315°, 67.5°, etc. | J |
| WSPM | Wind speed in m/s. | K |
| PM2.5 | Concentration of PM2.5 in $\mu g/m^3$. Used as the label/predicted value. | L |

Table 6: A summary of the statistical properties of the missingness of each Air Quality dataset. These datasets have fairly consistent missingness with around 1%-2% total missingness in each. The shading simply indicates a high/med/low within the table, with dark grey cells being the largest 1/3 for the column, and white being lowest 1/3.

| Dataset Air Quality | Miss % | Gap % | Structural % | Num Gaps $\geq 24$ | Gap (mean\|med\|max) |
|---|---|---|---|---|---|
| Aotizhongxin | 1.73% (7271) | 80.69% (5867) | 60.02% (4364) | 36 | 9.6\|2\|343 |
| Changping | 1.23% (5166) | 76.81% (3968) | 51.43% (2657) | 27 | 6.7\|3\|175 |
| Dingling | 1.67% (7015) | 81.67% (5729) | 46.03% (3229) | 24 | 8.2\|3\|1051 |
| Dongsi | 1.81% (7600) | 87.92% (6682) | 38.97% (2962) | 30 | 12.6\|2\|1517 |
| Guanyuan | 1.25% (5279) | 83.08% (4386) | 54.90% (2898) | 31 | 7.7\|3\|441 |
| Gucheng | 1.12% (4728) | 78.24% (3699) | 45.37% (2145) | 19 | 7.1\|3\|349 |
| Huairou | 1.78% (7485) | 75.50% (5651) | 55.83% (4179) | 41 | 6.8\|2\|211 |
| Nongzhanguan | 0.97% (4090) | 76.23% (3118) | 49.29% (2016) | 15 | 5.5\|2\|147 |
| Shunyi | 2.03% (8523) | 74.35% (6337) | 44.23% (3770) | 35 | 6.5\|2\|749 |
| Tiantan | 1.25% (5277) | 67.97% (3587) | 45.71% (2412) | 12 | 5.7\|3\|457 |
| Wanliu | 1.53% (6447) | 73.68% (4750) | 32.50% (2095) | 15 | 9.3\|2\|642 |
| Wanshouxigong | 1.22% (5146) | 81.56% (4197) | 59.41% (3057) | 27 | 7.5\|3\|93 |

Table 7: Min-Std buffer diagnostics for Air Quality. "Below train min" counts observed evaluation values below $x_{\min}^{train}$. Sentinel exceedance counts observed evaluation values below $x_{\min}^{train} - k\sigma^{train}$. $k_{\mathrm{req,max}}$ is the maximum train-standard-deviation buffer required to make sentinel exceedance zero across the analyzed variables for that split. Each $k$ cell reports total sentinel exceedance percentage, count, worst variable/fold exceedance percentage, and median Min-Max-space gap $g_k$. The Original Val+Test Holdout row combines the original validation and test splits for this diagnostic summary. Shading marks nonzero sentinel exceedances, with darker cells indicating larger total exceedance percentages.

| Split | Observed | Below train min | $k_{\mathrm{req,max}}$ | $k = 0.25$ | $k = 0.50$ | $k = 1.00$ | $k = 2.00$ |
|---|---|---|---|---|---|---|---|
| Train 5-fold Inner CV | 2,902,199 | 0.1393% (4,044) | 0.902 | 0.0432% (1,253) max 2.689% $g$ 3.27% | 0.0100% (291) max 1.271% $g$ 6.33% | 0.0000% (0) max 0.000% $g$ 11.91% | 0.0000% (0) max 0.000% $g$ 21.28% |
| Original Val+Test Holdout | 1,497,017 | 0.0546% (817) | 0.531 | 0.0208% (311) max 0.723% $g$ 3.04% | 0.0005% (7) max 0.038% $g$ 5.91% | 0.0000% (0) max 0.000% $g$ 11.15% | 0.0000% (0) max 0.000% $g$ 20.07% |

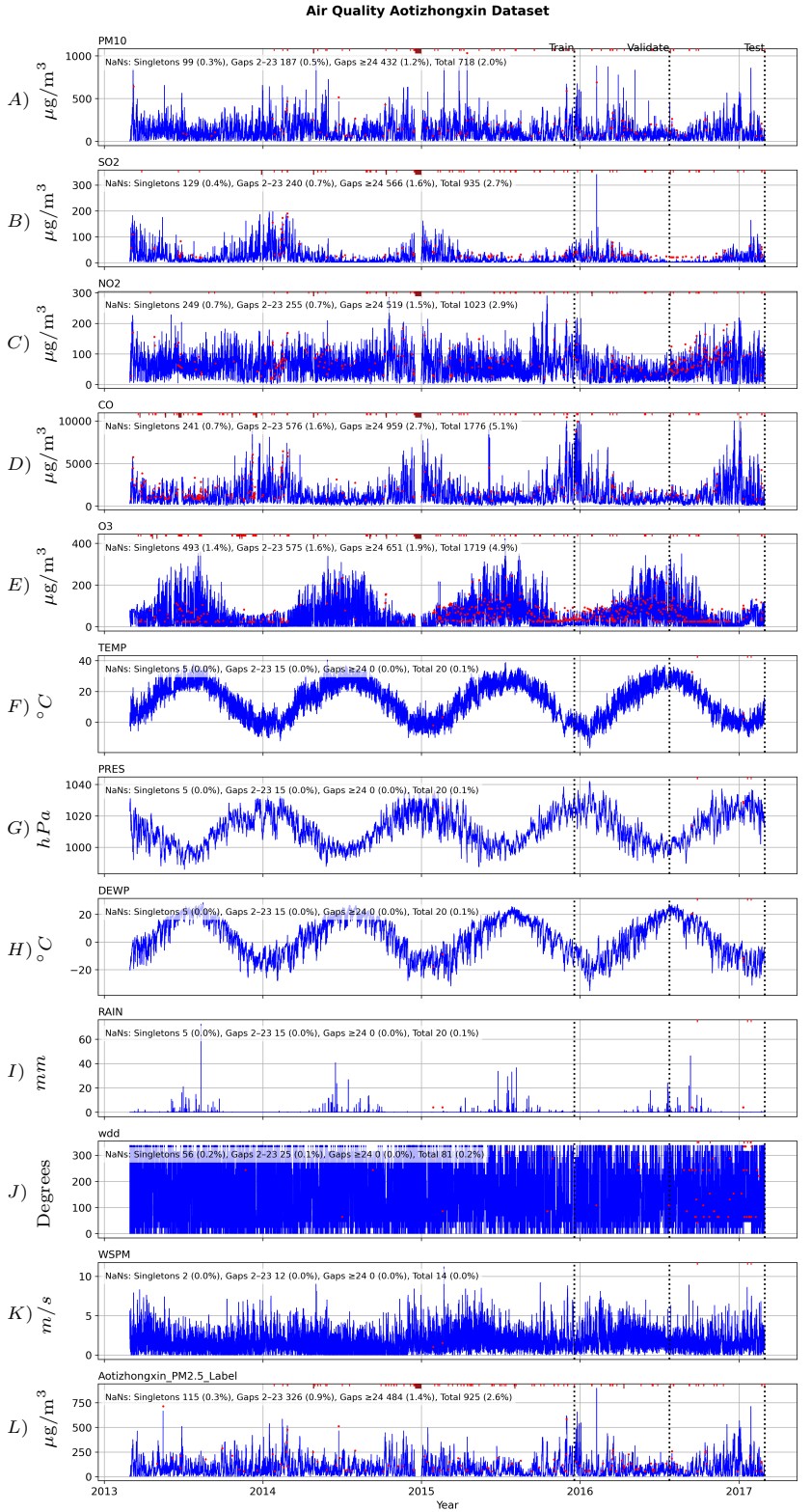

Figure 5: This plot shows the Aotizhongxin Air Quality dataset, the bottommost graph 'Aotizhongxin_PM2.5_Label' are the values being predicted. Red points on the graph indicate the presence of a singleton NaN, the red bars on the top indicate gaps in the data, with the larger darker bars being $\geq 24h$. The per-panel "NaNs:" line reports percentages relative to all timestamps in view (not to the NaN set). The plot also indicates the train/validation/test splits, with train start 2013-03-01 00:00:00, train end 2015-12-18 16:00:00, validation end: 2016-07-24 20:00:00, testing end: 2017-02-28 23:00:00. A total of 35064 time steps. The DateTime statistics are not included in this plot.

## C   Imputation Method Details

- **Constant Imputation**: These methods impute a constant value at every missing position, this may be called a "sentinel" or "flag" imputation. Some of these can be used as indicators that values are missing, and are not intended to attempt to find similar or "correct" values.

  - **Zero Imputation**: Replaces all missing values with zero. This is often inappropriate for approximation unless zero is a meaningful default or baseline. However it can work to flag that a value was missing. Note: after the downstream transformation the imputed value may not remain the same, and thus not work well as a NaN indicator post transformation; this is especially true if the variable contains both positive and negative values. For example, replacing missing values with 0 for a feature with range [-20,35] then applying a Min-Max transformation will result in a sentinel value that is not 0. This new sentinel value being unknown is not useful for indicating missing values.
  - **Mean Imputation**: Each missing value is filled using the mean of the corresponding variable. Note: this method does not allow for the imputed value to function as a NaN indicator, the imputed values will be difficult or impossible to differentiate from actual observed values.

- **Smoothing**: These methods propagate existing values or smooth functions but are not predictive of the original missing value:

  - **Forward Fill (ffill) then Backward Fill**: Missing values are replaced using the most recent non-missing value prior in time, followed by backward filling if any remain. The backfill will only be applied if the first time-steps in the variable are missing (meaning there are no non-missing prior values) and only applied until the first non-missing value is encountered.
  - **Rolling Mean Imputation**: Computes a moving average over a specified window, in our tests we selected a window size of 12 time-steps. Missing values are filled using this mean of recent history.

- **Model-based Imputation**: These methods try to infer the most probable true value for each missing position. The fitting process of these data dependent methods is more substantial and may be quite computationally expensive (see Table 1).

  - **K-Nearest Neighbors (KNN)**: Each missing value in each variable is replaced with the unweighted mean of the variable among the $k$ most similar rows (we use $k = 5$ and Euclidean distance similarity measurement) (Khayati et al., 2020). For a variable $x_j$ with a missing value at time $t$, let $\mathcal{N}_k(t)$ be the $k$ nearest/most similar rows (without a missing value in $x_j$); we impute with the mean of $x_j$ over $\mathcal{N}_k(t)$. The sklearn (Pedregosa et al., 2011) KNNImputer was used for testing with $k = 5$.
  - **Multiple Imputation by Chained Equations (MICE)**: Estimates/predicts the missing values in a variable using a regularized linear regression model based on other variables (Van Buuren & Groothuis-Oudshoorn, 2011). The missing values in a variable are predicted (using all other variables as predictors) one variable at a time; a model is fit on rows without missing values, and that model is used to fill rows in the variable that are missing. This process is repeated for each variable in a round robin fashion (variable with least missing first). Multiple iterations of this round robin occur (with subsequent passes using variables containing the imputed values to improve prediction) until either a maximum number of iterations is met, or convergence (meaning the largest absolute normalized change in an imputed value is less than a tolerance value).
  Using other variables to fit the current variable allows the method to model complex inter-variable relationships (Luo, 2022). The sklearn IterativeImputer was used for testing with default settings.
  - **MissForest**: A non-parametric, ensemble-based imputation approach using Random Forest regressors iteratively trained on non-missing data (Stekhoven & Bühlmann, 2012). This method works similar to MICE however the Random Forest regressor is used in place of regularized linear regression model. MissForest handles non-linear relationships and mixed data

types without distributional assumptions. For testing the sklearn RandomForestRegressor (with n_estimators=30, max_depth=4, max_features='sqrt', min_samples_leaf=5, and random_state=0) is used as the estimator for the sklearn IterativeImputer (with max_iter=2, and skip_complete=True to improve imputation times).

– **Kalman Smoothing/Filtering**: Each variable is processed independently, producing a 'smoothed' estimate of the variable. Missing values in the original variable are imputed with its counterpart from the smoothed. Kalman smoothing uses a linear-Gaussian state-space model to estimate missing values, recursively updating state estimates using observed data and model uncertainty (Kalman, 1960). It is effective for imputation on time series and short gaps, as it incorporates temporal dynamics and noise covariance. The pykalman KalmanFilter was used for testing with up to 5 EM iterations to estimate process and observation noise covariances.

KNN, MICE, and MissForest assign values in one variable by considering the values of multiple other variables simultaneously. This means they may infer structure between variables, while Kalman considers only the variable being imputed.

## D   Transformation Details

We evaluate the following transformation methods (using sklearn implementations):

- **Standard Scaling**: Each variable is standardized to have zero mean and unit variance:

$$x' = \frac{x - \mu}{\sigma}$$

  where $\mu$ and $\sigma$ are the mean and standard deviation, respectively.

- **Min-Max Scaling**: Maps each variable to the $[0, 1]$ interval:

$$x' = \frac{x - x_{\min}}{x_{\max} - x_{\min}}$$

- **Robust Scaling**: Standardizes each variable using the median and interquartile range (IQR) of the variable instead of mean and variance, making it more robust to outliers:

$$x' = \frac{x - \text{median}(x)}{\text{IQR}(x)}$$

- **Power Transform**: Applies a power-based transformation to make the data more Gaussian-like, followed by standardization. We use the Yeo-Johnson transform (Yeo & Johnson, 2000) as it works with both positive and negative values, and our datasets all include temperature values that can be negative. For a given parameter $\lambda$, the Yeo–Johnson transform $T(\lambda, x)$ is

$$T(\lambda, x) = \begin{cases} \{(x + 1)^{\lambda} - 1\}/\lambda, & x \geq 0, \ \lambda \neq 0, \\ \log(x + 1), & x \geq 0, \ \lambda = 0, \\ -\{(-x + 1)^{2-\lambda} - 1\}/(2 - \lambda), & x < 0, \ \lambda \neq 2, \\ -\log(-x + 1), & x < 0, \ \lambda = 2. \end{cases}$$

  After applying $T(\lambda, \cdot)$ to each variable, sklearn Power Transform standardizes the transformed values to zero mean and unit variance:

$$x' = \frac{T(\lambda, x) - \mu_{\lambda}}{\sigma_{\lambda}},$$

  where $\mu_{\lambda}$ and $\sigma_{\lambda}$ are the mean and standard deviation of the transformed training data.

- **Quantile Transform**: Maps each variable to a uniform distribution on $[0, 1]$ using its empirical cumulative distribution function. This can help with highly skewed distributions, and reduces the impact of outliers. For each variable with training samples $x_1, \ldots, x_n$, $x$ is mapped to the empirical cumulative distribution function:

$$x' = \hat{F}(x) = \frac{1}{n} \sum_{i=1}^{n} \mathbf{1}\{x_i \leq x\},$$

- **Max-Abs Scaling**: Maps each variable to the $[-1, 1]$ range based on its maximum absolute training value $\max_{1 \leq i \leq n} |x_i|$:

$$x' = \frac{x}{max_{1 \leq i \leq n}|x_i|}.$$

# E   Model Performance Metrics Details

**Notation:** We define $X_o(i)$ as the observed label value at timestep $i$, $X_p(i)$ as the predicted value at timestep $i$, $\bar{X}_o$ and $\bar{X}_p$ as the mean observed and predicted values respectively (over the testing set label), and $\sigma_{X_o}$ and $\sigma_{X_p}$ as the standard deviations of observed and predicted values respectively.

## E.1   Kling-Gupta Efficiency (KGE)

The Kling-Gupta Efficiency (KGE) is a metric used in hydrology to assess the predictive skill of a model. KGE refines its progenitor metric, the Nash-Sutcliffe Efficiency (NSE) (Nash & Sutcliffe, 1970), by decomposing error into correlation, variability, and bias:

$$\text{KGE} = 1 - \sqrt{(r-1)^2 + (\alpha - 1)^2 + (\beta - 1)^2} \tag{2}$$

where $r$ is the Pearson correlation (Pearson, 1895), $\alpha = \frac{\sigma_{X_p}}{\sigma_{X_o}}$ which represents variability bias as the ratio of standard deviations, and $\beta = \frac{\bar{X}_p}{\bar{X}_o}$ represents mean bias.

*Interpretation*: Based on Kling et al. (2012); Knoben et al. (2019):

- $\text{KGE} = 1$ : Perfect, the observed and predicted values are identical.

- $0.5 \leq \text{KGE} < 1$ : The predicted values do not exactly match the observed, but they are close. This is considered a good performance.

- $0 \leq \text{KGE} < 0.5$ : These results are considered poor, the predicted values do not match the shape of the observed.

- $\text{KGE} \leq 0$ : Often misinterpreted as a mean predictor score (as below) (Knoben et al., 2019). We will consider these results as Poor.

- $\text{KGE} \leq 1 - \sqrt{2} \approx -0.41$ : The predicted results are equal or worse than a mean predictor (Knoben et al., 2019) (that is, a model which predicts the mean label value for every timestep would result in equal/better performance).

## E.2   Normalized Mean Squared Error (NMSE)

The Normalized Mean Squared Error (NMSE) normalizes the MSE by the variance of observed values:

$$\text{NMSE} = \frac{\sum_{i=1}^{T}(X_o(i) - X_p(i))^2}{\sum_{i=1}^{T}(X_o(i) - \bar{X}_o)^2} \tag{3}$$

By normalizing the MSE the resulting value is scale-invariant and has no physical units; this allows comparison between results from distinct datasets with different ranges. While many similar studies will use the MSE to measure performance, the resulting MSE values between studies (using different data or testing splits) are not comparable. Because NMSE is a constant multiple of MSE it preserves all information contained in MSE for performance ranking, while also being more informative when comparing across studies.

Note that using this definition of NMSE we have $\text{NMSE} = 1 - \text{NSE}$. This is useful for formulating the following interpretation using the performance evaluation criterion of NSE.

*Interpretation*: per suggestions from Moriasi et al. (2015):

- $\text{NMSE} = 0$ : Perfect, the observed and predicted values are identical.

- $\text{NMSE} < 0.2$ : Very Good.

- $\text{NMSE} < 0.3$ : Good.

- NMSE $< 0.5$ : Satisfactory.

- $0.5 <$ NMSE $< 1$ : Not Satisfactory.

- NMSE $= 1$ : Model is a mean predictor.

- NMSE $> 1$ : The predicted results are worse than a mean predictor, the errors exceed data variance.

### E.3 Threshold F1 Score

To assess the model's temporal accuracy and ability to detect extreme events, we use a quantile-based thresholding approach. We define a *peak* as a label value above a specified threshold $\tau$ (set at the $99^{th}$ percentile of the observed series). Each timestep is treated as a binary classification: peak or not peak.

We compute the **Peak Hit Rate** (PHR), the proportion of correctly predicted peaks; and the **Peak False Discovery Rate** (PFDR), the proportion of false positives.

$$\text{PHR} = \frac{\sum_{i=1}^{T} \mathbf{1}(X_o(i) \geq \tau \text{ and } X_p(i) \geq \tau)}{\sum_{i=1}^{T} \mathbf{1}(X_o(i) \geq \tau)}$$

$$\text{PFDR} = \frac{\sum_{i=1}^{T} \mathbf{1}(X_o(i) < \tau \text{ and } X_p(i) \geq \tau)}{\sum_{i=1}^{T} \mathbf{1}(X_p(i) \geq \tau)}$$

$$\text{precision} = \frac{\sum_{i=1}^{T} \mathbf{1}(X_o(i) \geq \tau \text{ and } X_p(i) \geq \tau)}{\sum_{i=1}^{T} \mathbf{1}(X_p(i) \geq \tau)} = 1 - \text{PFDR}$$

This formulation converts the continuous time-series regression into a binary extreme event classification problem. PHR corresponds to recall or sensitivity as in ROC (receiver operating characteristic) curve analysis (Fawcett, 2006), while PFDR corresponds to $1 - \text{precision}$ (Davis & Goadrich, 2006; Saito & Rehmsmeier, 2015). We combine these into a single **F1 Score (99th)**, defined as:

$$\text{F1 Score (99th)} = \frac{2 \cdot \text{PHR} \cdot \text{precision}}{\text{PHR} + \text{precision}}$$

If no predicted peaks occur we set F1 (99th) $= 0$.

This rewards high detection rates, and low false discovery rates, giving a balanced measure of extreme event detection.

### E.4 Composite Score

To reduce result complexity and enable model comparison, we construct a single multi-criteria "Composite Indicator" (Greco et al., 2019; Nardo et al., 2005) which integrates the three core metrics: shape accuracy (KGE), magnitude accuracy (NMSE), and extreme event accuracy (F1 Score at the $99^{\text{th}}$ percentile).

To construct our Composite Score, we adopt the composite indicator methodology outlined in the The Organisation for Economic Co-operation and Development (OECD) *Handbook on Constructing Composite Indicators* (Nardo et al., 2005). The OECD handbook is a comprehensive and widely accepted technical guide for the design of composite metrics, offering robust and generalizable procedures applicable across disciplines—including economics, environmental science, education, and beyond.

We use equal weighting for a composite score on the clipped metric values (the mean of the individual metrics). While Greco et al. (2019); Nardo et al. (2005) warn that equal weighting implies compensability (i.e., a poor result in one metric can be fully offset by good scores in another), the strategy of clipping results should mitigate this by preventing any single metric from dominating the score. The principled case for equal weighting here is transparency and interpretability. Equal weighting avoids data-dependent weights that would differ by dataset, units/data scale that would hinder cross-domain comparisons. If we were to

choose a data driven weighting scheme (PCA), we would need separate scores for separate datasets. For this reason we choose a simple equal weighting which enables interpretability and comparability across models using different datasets.

## F   Additional Results Plots

This appendix provides a detailed distributional analysis of prediction scores for a representative selection of preprocessing strategies. To illustrate the variance in performance, Figures 6 - 9 contrast the top-performing method (Min-Std | Min-Max) against other common approaches at timestep t+4.

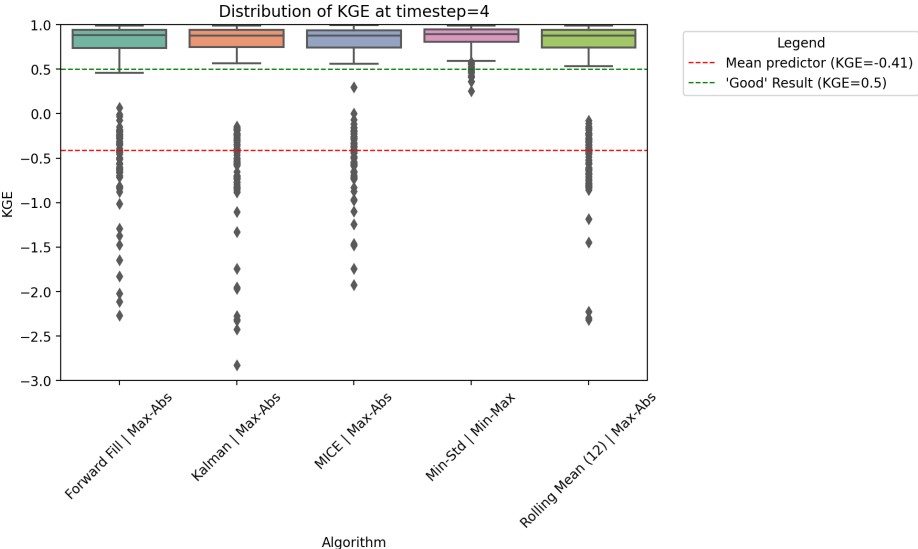

Figure 6: Distribution of Kling-Gupta Efficiency (KGE) scores at prediction horizon $t + 4$. The dashed green line indicates the generally accepted threshold for 'Good' predictive performance (KGE $\geq 0.5$), while the dashed red line marks the performance of a mean predictor (KGE $= -0.41$). The Min-Std | Min-Max method yields the fewest severe outliers falling below the mean predictor threshold, demonstrating highly robust shape accuracy across diverse environmental data.

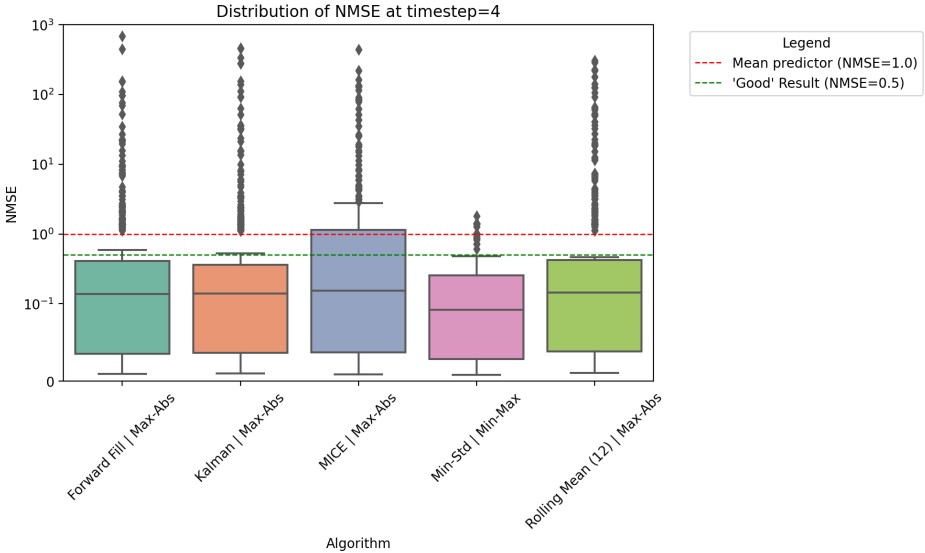

Figure 7: Distribution of Normalized Mean Squared Error (NMSE) at prediction horizon $t + 4$, displayed on a logarithmic scale. The dashed green line indicates 'Good' error magnitudes (NMSE $\leq 0.5$) and the red dashed line denotes a mean predictor (NMSE $= 1.0$). NMSE shown on a symmetric log scale (linthresh=0.1) with limits [0, 1000]

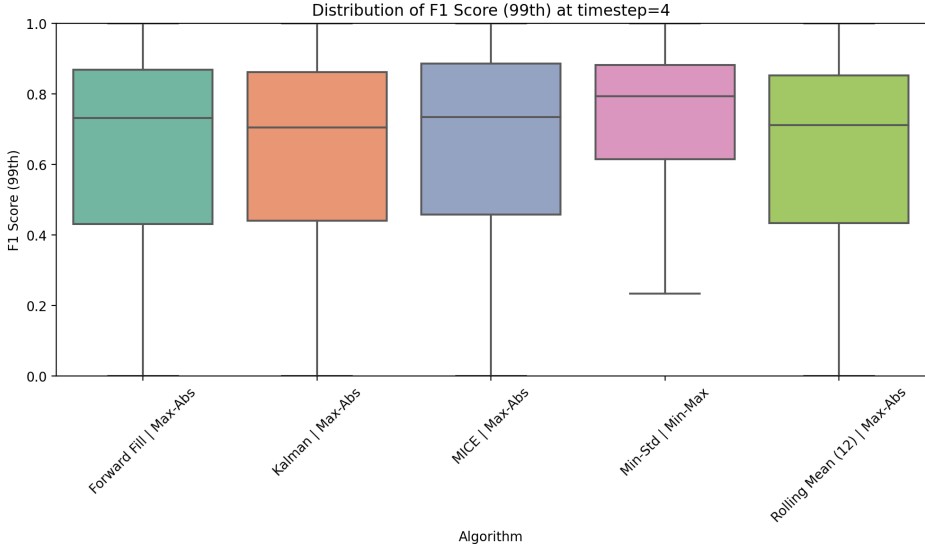

Figure 8: Distribution of the thresholded F1 Score ($99^{th}$ percentile) at prediction horizon $t + 4$. This metric evaluates the models' capacity to correctly predict extreme temporal events. The Min-Std | Min-Max configuration demonstrates the highest median detection accuracy and maintains a tighter distribution of positive outcomes than comparative methods.

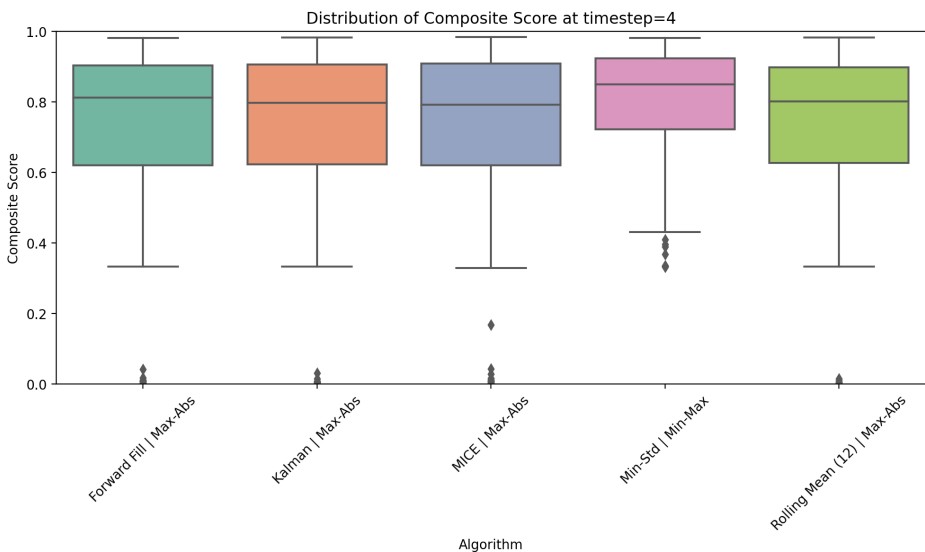

Figure 9: Distribution of Composite Scores at prediction horizon $t + 4$. Each boxplot represents $n = 140$ independent evaluations (28 datasets $\times$ 5 random initializations). The proposed Min-Std | Min-Max preprocessing strategy achieves the highest median composite score and the narrowest interquartile range, indicating superior and more consistent overall performance compared to standard baselines.

Figures 10 and 11 show Cirical Difference diagrams similar to Figure 3, but considering dataset-families separately (i.e. only GRWS results or only Air Quality results). Because splitting the benchmark reduces the number of blocks from $N = 112$ to $N = 48$ and $N = 64$, the post-hoc tests have less power and should not be interpreted as replacing the combined analysis. Nevertheless, Min-Std | Min-Max remains the best-ranked method in both dataset families, indicating that the combined result is not driven solely by one dataset family.

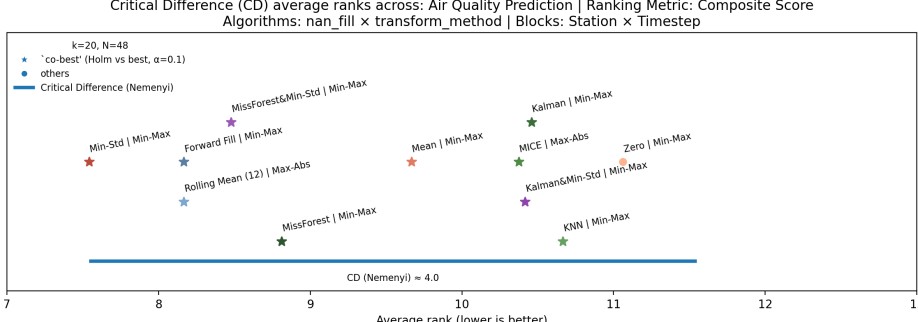

Figure 10: Dataset-family sensitivity analysis. Critical-difference diagram for Air Quality datasets using the same ranking procedure as Figure 3.

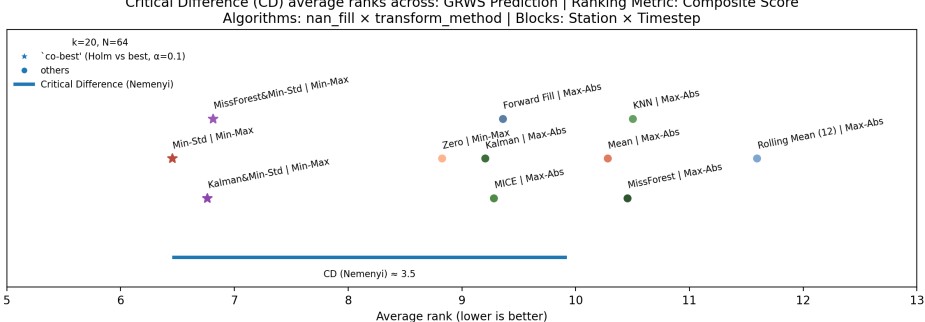

Figure 11: Dataset-family sensitivity analysis. Critical-difference diagram for GRWS datasets using the same ranking procedure as Figure 3.

