# OpenReview forum: "Learning from Missing Values: Encoding Missingness in Representation-Space for LSTM Time Series Forecasting"
_TMLR — Under review for TMLR_

### Review · Reviewer_N8BS · 2026-03-20

**Summary Of Contributions:**

- introduction of a new missingness encoding type (Minimum minus Std.Dev.)
- extensive experiments to compare to existing methods (12 of them), on various datasets (28 time series), and 6 possible data pre-processing (scaling etc)
- an open dataset and the corresponding metrics/detailed workflow (=a standardised benchmark)

**Additional Comments:**

Sorry for being late..

**Audience:**

Yes

**Audience Explanation:**

Yes, all people interested in real world applications have to deal with missing values, in time series or elsewhere, and this method could be useful (for tabular data too actually, not necessarily only time series).

**Claims And Evidence:**

Yes

**Claims Explanation:**

Yes, apart from the Dropout argument, that I call to question. The debate can be concluded by running experiments.

**Requested Changes:**

## 1. vocab.

> imputation method ‘Min-Std’ presented in this paper

At this point, explicit already that Min-Std is "Minimum minus Std.dev." (and not Min dash Std), for clarity.


## 2. Dropout

There is some confusion on the Dropout analogy.
- In the introduction, there is some mention of Dropout acting on features, without the detail that the dropout considered is *input* Dropout. This is confusing, as it leads to think that the paper is confusing inputs fed to the LSTM and activations within the LSTM. Please mention *input* dropout earlier in the manuscript, to avoid this confusion, if this is really what is done.
- but then, in sec 3.1.1, I do not see any input dropout used, in the architecture, it's actually in between LSTM layers that Dropout is used. This fact challenges the interpretation made in the earlier sections, stating that turning NaNs into 0's (after inputation+transform) is somewhat equivalent to (input) dropout.

So, either:
- correct the architecture in 3.1.1, adding input dropout, and mention the "input" keyword early on, if it's what was done.
- or, (most likely), correct the motivation in the introduction. The input values (attributes) coming from the data (and that are imputed with Min-Max here) do not have the same intrinsic meaning as features (embeddings) living inside the networks (values of the activations coming out of neurons). Dropout sets activities to 0 because 0, for most activation function, is a natural "no activity", neutral state, that just decreases the sum of activities received by the next neuron. In contrast, input values (dataset attributes, also called features sometimes) have no reference 0. After Min-Max scaling, 0 is the smallest possible value, but is not "neutral" either.

In summary, the following sentences have to be corrected:
- "the only 0 values the model will receive are either missing values, or values dropped by a dropout regularizer." (abstract)
- "is to encode missing values as the same 0",
- "the imputed values being the same as values dropped in the dropout layer of the LSTM",
- "treat both imputed missing values, and dropout masked values as uninformative,"

If authors do not agree, it is easy to settle the argument:
- either include an input Dropout in the backbone, see if it changes anything (not sharpest experiment, since various results may be explained invarious ways)
- try the Max+Std imputation method. If it's equivalent to Min-Std, then I'm correct. If it's much worse, then I was wrong (and this is very interesting!)


## 3. Max+Std ?

Still related: Why is 0 (the lowest possible value) a good choice, and not instead the highest possible value? In particular, since Min-Max scaling is coming out as the best transform (MAx-Abs also shares the same decisive property), why not consider Max+Std as a possible alternative ? It would also appear clearly as a flag to the network, equivalently to Min-Std. I do not request such an experiment, but would be curious to see the results of it. If they are equivalent, it invalidates the Dropout explanation, and strengthens the "flag sent to NN" interpretation. If Max+Std performs worse, then it means 0/1 are not symmetric, which after all is possible in min-max scaling (or max-abs if inputs are mostly positive) when it is followed by ReLU and other non-symmetric functions.


## 4. k factor

About the k (buffer) factor: I think it would be rather easy to check, even only on the train set, how much it is likely to have values exceeding the buffer:
Doing cross-validation (split in train/val), one could look at the percentage of  validation examples that are beyond the Min-Std "fitted" on the train set, and average over folds. I guess a good k is one where such events are extremely rare. An extra buffer that helps the network really flag NaNs as "out of domain" may actually also help the LSTM. This would be interesting, as it would help understand the origin of the success of the Min-Std imputation: it is enough to have no test value exceeding the threshold, or it it better that NaNs be far away form actual values ?

## 5. Number of experiments?
Could you explicit the number of experiments that are summarized in the box plot of fig 1 and following figures? I read N=112 for the Air quality dataset, nothing for GRWS. N=112 comes from (#time series)x(#target values)x(#time horizons) ? It's important to explicit it, it'd help understand the manuscript altogether.

## 6. Additional plot

Although I appreciate a lot the effort made in statistical analysis to summarize the (numerous) experiments made, it is slightly frustrating to not have a close-up look at the best vs typical algorithms, in particular, the aggregated metric (composite score) hides the details of how much each algorithm is biased in a way or the other. Could you report in appendix, a *slightly* more detailed view, for instance, for a few inputations (Min-Std|Min-Max, Forward-Fill|Max-Abs, Rolling-Mean|Max-Abs, +1 more), provide for each, the distribution of score (1 distrib per score, scores are F1-99, KGE and NMSE) ? Each distribution coming from your numerous experiments (1 point of density in the distrib = 1 experiment (datasets x time horizons)). Or maybe just consider horizon t+4 ?

## typo:
- notable faster -> notably faster
- ? the most probably true value -> the most probable value ?

---

> ### Comment · Reviewer_N8BS · 2026-05-20
> **Answers from authors ?**
>
> I believe answers and resubmission is expected from the authors, before a recommendation can be made.

---

> ### Author Response · Authors · 2026-05-22
> **Responses to Reviewer 1 (N8BS)**
>
> **Note:** There is a “Common Themes” comment on the paper that addresses comments that are similar across reviewer comments. It is intended to reduce repetition, and will be referenced within individual responses.
>
> ## Vocab.
>
> imputation method ‘Min-Std’ presented in this paper
>
> At this point, explicit already that Min-Std is "Minimum minus Std.dev." (and not Min dash Std), for clarity.
>
> **Response:** This is a great idea.
>
> **Change Made:** Added to first mention in intro.
>
> ## Input Dropout
>
> There is some confusion on the Dropout analogy.
>
> * In the introduction, there is some mention of Dropout acting on features, without the detail that the dropout considered is *input* Dropout. This is confusing, as it leads to think that the paper is confusing inputs fed to the LSTM and activations within the LSTM. Please mention *input* dropout earlier in the manuscript, to avoid this confusion, if this is really what is done.
> * but then, in sec 3.1.1, I do not see any input dropout used, in the architecture, it's actually in between LSTM layers that Dropout is used. This fact challenges the interpretation made in the earlier sections, stating that turning NaNs into 0's (after inputation+transform) is somewhat equivalent to (input) dropout.
>
> So, either:
>
> * correct the architecture in 3.1.1, adding input dropout, and mention the "input" keyword early on, if it's what was done.
> * or, (most likely), correct the motivation in the introduction. The input values (attributes) coming from the data (and that are imputed with Min-Max here) do not have the same intrinsic meaning as features (embeddings) living inside the networks (values of the activations coming out of neurons). Dropout sets activities to 0 because 0, for most activation function, is a natural "no activity", neutral state, that just decreases the sum of activities received by the next neuron. In contrast, input values (dataset attributes, also called features sometimes) have no reference 0\. After Min-Max scaling, 0 is the smallest possible value, but is not "neutral" either.
>
> **Response:** See ‘Common Themes’ comment section ‘4) Dropout’ for response.
>
> **Change Made:** The sentences noted by the reviewer have been changed to show they are inspired/motivated by dropout, and not implying that input dropout is used in this study.
>
> * try the Max+Std imputation method. If it's equivalent to Min-Std, then I'm correct. If it's much worse, then I was wrong (and this is very interesting\!)
>
> ## Max+Std ?
>
> Still related: Why is 0 (the lowest possible value) a good choice, and not instead the highest possible value? In particular, since Min-Max scaling is coming out as the best transform (MAx-Abs also shares the same decisive property), why not consider Max+Std as a possible alternative ? It would also appear clearly as a flag to the network, equivalently to Min-Std. I do not request such an experiment, but would be curious to see the results of it. If they are equivalent, it invalidates the Dropout explanation, and strengthens the "flag sent to NN" interpretation. If Max+Std performs worse, then it means 0/1 are not symmetric, which after all is possible in min-max scaling (or max-abs if inputs are mostly positive) when it is followed by ReLU and other non-symmetric functions.
>
> **Response:**  Future work is to consider Min-Std as a member a class of “Representation-Space Missingness Encoders”, upper bound sentinels such as max+std will be considered, as well as potentially more robust alternatives like Min-MAD. However, for this study with LSTM and environmental data a lower bound was selected; given the dataset features and labels, high values are generally important extremes, and low values are less impactful. Given this, we consider that if val/test data has any out-of-distribution (OOD) values they are likely on the high-end, thus an upper bound sentinel method is more likely to have OOD values approaching the sentinel. Also, as noted above, when considering a sentinel value we looked to dropout
>
> To your point, I believe that Max+Std (or any upper bound sentinel) may have an additional benefit; because upper bound sentinels force the observed values to occupy the lower part of \[0,1\], and IEEE-style binary floats are more densely spaced near zero than near one, upper bound sentinels may preserve slightly more absolute numerical resolution for the real observed values after scaling. This is likely negligible with float32, but may be worth testing with float16/mixed precision.
>
> **Change Made:** Expanded final “Future Work” paragraph in conclusion.

---

> > ### Author Response · Authors · 2026-05-22
> > **Responses to Reviewer 1 (N8BS) (continued)**
> >
> > ## k factor
> >
> > About the k (buffer) factor: I think it would be rather easy to check, even only on the train set, how much it is likely to have values exceeding the buffer: Doing cross-validation (split in train/val), one could look at the percentage of  validation examples that are beyond the Min-Std "fitted" on the train set, and average over folds. I guess a good k is one where such events are extremely rare. An extra buffer that helps the network really flag NaNs as "out of domain" may actually also help the LSTM. This would be interesting, as it would help understand the origin of the success of the Min-Std imputation: it is enough to have no test value exceeding the threshold, or it it better that NaNs be far away form actual values ?
> >
> > **Response:** This is a great addition, we have run this analysis which also confirms the k=1.0 choice. We performed this on cross-validation (5 folds in training set) as requested, and also using the original split fitting min and std on 70% train.
> >
> > See ‘Common Themes’ comment Section 1\) K-factor for further response.
> >
> > ## Number of experiments?
> >
> > Could you explicit the number of experiments that are summarized in the box plot of fig 1 and following figures? I read N=112 for the Air quality dataset, nothing for GRWS. N=112 comes from (\#time series)x(\#target values)x(\#time horizons) ? It's important to explicit it, it'd help understand the manuscript altogether.
> >
> > **Response:** See ‘Common Themes’ comment Section 3\) Number of experiments for response.
> >
> > ## Additional plot
> >
> > Although I appreciate a lot the effort made in statistical analysis to summarize the (numerous) experiments made, it is slightly frustrating to not have a close-up look at the best vs typical algorithms, in particular, the aggregated metric (composite score) hides the details of how much each algorithm is biased in a way or the other. Could you report in appendix, a *slightly* more detailed view, for instance, for a few inputations (Min-Std|Min-Max, Forward-Fill|Max-Abs, Rolling-Mean|Max-Abs, \+1 more), provide for each, the distribution of score (1 distrib per score, scores are F1-99, KGE and NMSE) ? Each distribution coming from your numerous experiments (1 point of density in the distrib \= 1 experiment (datasets x time horizons)). Or maybe just consider horizon t+4 ?
> >
> > **Response:** Great suggestion, I selected the imputations you suggested as well as MICE and Kalman Smoothing.
> >
> > **Change Made:** An “Additional Results Plots” appendix has been added containing four plots (one per performance metric plus composite metric). I also added a paragraph to the discussion section (3rd paragraph) regarding these plots and discussing the narrower IQR and fewer outliers of Min-Std compared to the requested imputations.
> >
> > ## typo:
> >
> > * notable faster \-\> notably faster
> > * ? the most probably true value \-\> the most probable value ?
> >
> > **Change Made:** Corrected.

---

### Review · Reviewer_AQCf · 2026-03-28

**Summary Of Contributions:**

The paper proposes a hybrid method of flagging missing data via imputation with outlier value. It follows the logic behind zero imputation but introducing a value that is very low while not overpowering making calculations unstable. The method is simple and elegant. They also test multiple preprocessing methods in combination. They also test this method in combination with other imputation methods.

**Audience:**

Yes

**Audience Explanation:**

The idea of an imputation that serves as a flag is simple but also interesting as it can be similar to the flagging method but with half the inputs. So if it works, it would be of great interest.

**Claims And Evidence:**

No

**Claims Explanation:**

The paper suffers from too many issues at its current form.

Major:
- The paper is unnecessarily long and seeing the note that this is shortened already is very surprising. As there is a lot of unnecessary details in this writing that it makes the paper unreadable. Lots of explanation of metrics meaning and datasets which are already referenced in other publications. And lots of rhetoric on the value of the method rather than showing evidence.
- That being said, the paper also lacks appropriate testing or mathematical proof to convince me that the approach they propose is as powerful as they claim it to be. There is no mathematical or experimental work that leads them to make the decision on the mean - 2 * std. Why this particular value? Have they systematically tried other values to reach this? I only saw mean - 1* std. They also talk about zero imputation but do not test it even though it is actually a very similar approach to theirs.
- There are also many powerful methods that have not been tested or even mentioned such as imputation + flagging, modulation layer (https://openreview.net/forum?id=MRLHN4MSmA), GRAPE (https://arxiv.org/abs/2010.16418), VAEAC (https://openreview.net/forum?id=SyxtJh0qYm), Neumiss (https://arxiv.org/abs/2007.01627).
- In figure 1, I am not sure of what model is being tested in the preprocessing method filteration here. Is it an average of all imputation/flagging methods? If you already have that, then why not using all combinations in the subsequent analyses. There might be certain methods that combined with certain preprocessing methods yield better performance. It's also visible from the error bars that there is a  very high variability meaning it is hard to say the top two methods are significantly better than the others. Being on average better does not necessarily mean they are the best as I mentioned above, there could be combinations that they excel in and are poor at others.
- Given the authors are testing on two datasets only, I am wondering where the N=112 at the top of page 10 is. Also, why not present performance on each dataset? The too much processing that is being plotted there is making me doubt a lot of the intermediate results.

With these problems and the current writing, it is very hard to actually evaluate the paper well.

**Requested Changes:**

The writing needs an overhaul removing many sections and explaining more results as mentioned above.

---

> ### Author Response · Authors · 2026-05-22
> **Responses to Reviewer 2 (AQCf)**
>
> **Note:** There is a “Common Themes” comment on the paper that addresses comments that are similar across reviewer comments. It is intended to reduce repetition, and will be referenced within individual responses.
>
> ## Length
>
> The paper is unnecessarily long and seeing the note that this is shortened already is very surprising. As there is a lot of unnecessary details in this writing that it makes the paper unreadable. Lots of explanation of metrics meaning and datasets which are already referenced in other publications. And lots of rhetoric on the value of the method rather than showing evidence.
>
> **Response:** We appreciate that many details are common and available elsewhere; however as the paper is intended for a broad audience, we chose to include a greater detail of information so that readers who may be less familiar with the background material than the reviewer would not need to look up secondary sources to understand the paper. The description of performance metrics is included to explain why these specific metrics are selected to create the composite score, these were not arbitrary choices and the section outlines their exact purpose in the score. We agree that the original manuscript contained standard background that could be shortened.
>
> **Change Made:** Removed Section 3.4 Process Summary, reduced introductory motivation, condensed baseline descriptions. Additional text has been added to provide details requested by other reviewers/comments, however the revised document is still within the required page limit.
>
> ## k-factor
>
> That being said, the paper also lacks appropriate testing or mathematical proof to convince me that the approach they propose is as powerful as they claim it to be. There is no mathematical or experimental work that leads them to make the decision on the mean \- 2 \* std. Why this particular value? Have they systematically tried other values to reach this? I only saw mean \- 1\* std.
>
> **Response:** The method described in this paper is the **minimum** \- k\*std, not **mean** \- k\*std. I have verified the original submission did not contain any accidental mention of “mean \- k\*std”.  I recognize that it is very common to discuss std in terms of mean, and this reasonable assumption may be due to the common connection of mean in tandem with std.  This distinction is vital, as the objective of the method is to have an extreme sentinel value that maps to exactly zero after transformation, imputing the min-std guarantees an extrema, while mean-std does not. In terms of which values of k were tested, we tested values 0.25, 0.5, and 1, this is noted in the last discussion paragraph, and second last paragraph in section 2.2.
>
> See ‘Common Themes’ comment Section 1\) K-factor for further response.
>
> **Change Made:** distinction of mean-std not producing an extrema included in introduction
>
> ## Zero Imputation
>
> They also talk about zero imputation but do not test it even though it is actually a very similar approach to theirs.
>
> **Response:** All imputation/transformation combinations described in S2 were tested, however Reviewer 2 is correct that Zero Imputation was not included in Figure 3\.
>
> I would also like to note that zero imputation is not as similar as you indicate. Consider temperature values ranging from \-20 to \+20; if a value is missing, zero imputation will assign it a value of 0 ( a value within the variable range, and a value that contains information). When scaled, that imputed value will change (to \~0.5) and remain within the variable range, it will not be possible to differentiate from values recorded at actual temperatures of 0 degrees and thus not function as a sentinel. However, with min-std that missing value will be imputed with a value around \-31, and when scaled with minmax it will become exactly 0, and no other non-missing value will be 0, ensuring it does function as an identifiable sentinel.
>
> We could also consider this with a strictly positive variable, such as precipitation. A value of 0 indicates an important piece of information ‘no rain’, missing recordings imputed as 0 would be impossible to differentiate from ‘no rain’. While using min-std we would have a unique missing flag sentinel value, with a ‘buffer’ between the sentinel value and the actual data.
>
> **Change Made:** Replaced Fig 3 with corrected version that does include Zero Imputation that was mistakenly omitted from the original plot.

---

> > ### Author Response · Authors · 2026-05-22
> > **Responses to Reviewer 2 (AQCf) (Conitued)**
> >
> > ## Additional Baseliness
> >
> > There are also many powerful methods that have not been tested or even mentioned such as imputation \+ flagging, modulation layer ([https://openreview.net/forum?id=MRLHN4MSmA](https://openreview.net/forum?id=MRLHN4MSmA)), GRAPE ([https://arxiv.org/abs/2010.16418](https://arxiv.org/abs/2010.16418)), VAEAC ([https://openreview.net/forum?id=SyxtJh0qYm](https://openreview.net/forum?id=SyxtJh0qYm)), Neumiss ([https://arxiv.org/abs/2007.01627](https://arxiv.org/abs/2007.01627)).
> >
> > **Response:** See ‘Common Themes’ comment Section 2\) Baseline Imputation Strategies for response.
> >
> > ## Filtering
> > In figure 1, I am not sure of what model is being tested in the preprocessing method filteration here. Is it an average of all imputation/flagging methods? If you already have that, then why not using all combinations in the subsequent analyses. There might be certain methods that combined with certain preprocessing methods yield better performance. It's also visible from the error bars that there is a very high variability meaning it is hard to say the top two methods are significantly better than the others. Being on average better does not necessarily mean they are the best as I mentioned above, there could be combinations that they excel in and are poor at others.
> >
> > **Response:** This filtering step is comparing the preprocessing methods,  plots each contain the results from a single preprocessing method and ALL imputation methods. The large error bars are expected as we have 28 different datasets with diverse missingness patterns, distributions, etc. Also each test is repeated 5 times with different seeds increasing variance across results.
> >
> > I agree that filtering based solely on the box plots would be ill advised, however the filtering was also based on the results of Figure 2 which ranks methods across each individual datasets at each timesteps separately which takes into account this variability in the 28 datasets. The null hypothesis shows there is a statistical difference, and both post-hoc tests clearly indicate that Min-Max and Max-Abs out perform the alternatives. Both figures indicate the same two methods performing better than the other four. This result is not surprising as the non-Gaussian distribution in data does not lend itself to all transformation methods equally.  This is discussed in the first discussion paragraph.
> >
> > ## Number of Experiments
> >
> > Given the authors are testing on two datasets only, I am wondering where the N=112 at the top of page 10 is.
> >
> > **Response:** See ‘Common Themes’ comment Section 3\) Number of experiments for response.
> >
> > ## Additional Plots
> >
> > Also, why not present performance on each dataset? The too much processing that is being plotted there is making me doubt a lot of the intermediate results.
> >
> > **Response:** The cited paper ‘*Statistical comparisons of classifiers over multiple data sets*’ by Demšar describes the Critical Difference Average Ranking and post-hoc tests used in this paper specifically for the comparison of multiple learning algorithm variants across multiple datasets. This is a standard reference (Google Scholar reporting over 17,000 citations and Semantic Scholar reporting over 12,000 citations) and the described methods are intended for comparisons such as in this paper. We agree that additional disaggregated views are useful, however providing 28 plots (one per dataset) is impractical. To address the concern that the pooled analysis may obscure behaviour, we include separate CD plots (similar to figure 3\) for ‘dataset-family’: Air Quality (N=48) and GRWS (N=64). As expected, the smaller N reduces post-hoc power, so these plots are not used to replace the combined analysis. However, Min-Std | Min-Max remains the best-ranked method in both dataset families, supporting that the combined result is not driven by only one dataset family.
> >
> > **Change Made:** An “Additional Results Plots” appendix has been added containing the two additional dataset-family CD plots. This appendix also contains box plots per performance metric per request of Reviewer 2 (AQCf). The per-metric view should also clarify intermediate results. A paragraph has been added to the discussion section (3rd paragraph) regarding these plots and discussing the narrower IQR and fewer outliers of Min-Std compared to the requested imputations.

---

### Review · Reviewer_x3TT · 2026-04-23

**Summary Of Contributions:**

The Authors use a specific token for missing values in a time that allows an "unknown" category in the classification.  This differs from training the system to estimate feasible values for the missing data points. This is not far away from systems that improve resolution in static data. You may fill it fit generated, most probable values, but has a problem in cases when something exceptional happens - the most probable selection may not reflect the actual situation.

The Manuscript provided provides an explicit solution  for filling the gaps, but at the same time, my the "NULL" data marking make sure that inference from only estimated, not measured  data points. There is certainly a benefit to make sure that the danger of modal collapse would happen when generated data starts to enforce its preference in the training.

Using Missingness concept, the Manuscripts describes that it reduces the overfitting and modal collapse and provides better results.

**Additional Comments:**

Sorry, to be so late in doing the review.

**Audience:**

Yes

**Audience Explanation:**

The Manuscript is well and clearly written, even it is not pointing on the reasons, I think are crucial to understanding why it works. The fact that it provided evidence that nullifying the NULLs is beneficial is strong enough of a step forward to make it interesting for the TMLR audience.

**Broader Impact Concerns:**

No Concerns.

**Claims And Evidence:**

Yes

**Claims Explanation:**

Avoiding the polluted effect of generated content in the training data by marking it clearly for the training process, I think, is the core of the Manuscript.  The experiments are supporting  the conclusion.  I think it is a convincing and clear evidence of the benefits the Missingness concept.

**Requested Changes:**

The Authors should comment on the role of modal collapse in generative filling of missing points.

---

> ### Comment · Reviewer_x3TT · 2026-05-14
> **Taking another look at the paper and considering the role of imputation in datasets in the light of the danger of using generated data to train neural networks.**
>
> I tend to think that  the quality of data matters in a way that the "final" models should use data where there are no missing data entries that have to be imputed.  Using the output of trained AI solutions to fill gaps have tendency to increase the probability of certain types of imputations, that may increase the probability of the inputed choices with respect to other entries leading to a distorted predictions of probabilities of the next entries.
>
> An excellent imputation method should still be the obvious one - do the measurements meticulously again to  avoid the missing record.  After all,  the science is fundamentally experimental and advances are created with better, high quality data.
>
> Currently I am missing the discussion about distorted imputations and modal collapse in the Manuscript.

---

> ### Author Response · Authors · 2026-05-22
> **Responses to Reviewer 3 (x3TT)**
>
> **Note:** There is a “Common Themes” comment on the paper that addresses comments that are similar across reviewer comments. It is intended to reduce repetition, and will be referenced within individual responses.
>
>
> **Modal Collapse**
>
> The Authors should comment on the role of modal collapse in generative filling of missing points.
>
> #### (Comment added May 13):
>
> #### Taking another look at the paper and considering the role of imputation in datasets in the light of the danger of using generated data to train neural networks.
>
> I tend to think that the quality of data matters in a way that the "final" models should use data where there are no missing data entries that have to be imputed. Using the output of trained AI solutions to fill gaps have tendency to increase the probability of certain types of imputations, that may increase the probability of the inputed choices with respect to other entries leading to a distorted predictions of probabilities of the next entries.
>
> **Response:** See ‘Common Themes’ comment Section 2\) Baseline Imputation Strategies for response.
>
> An excellent imputation method should still be the obvious one \- do the measurements meticulously again to avoid the missing record. After all, the science is fundamentally experimental and advances are created with better, high quality data.
>
> **Response:** We completely agree that the gold standard is to meticulously collect high-quality data without gaps. However, in the domain of retrospective environmental monitoring and remote-sensing, "repeating the measurement" is physically impossible. If a hydrological/meteorological gauge, or air-quality sensor fails during a historical storm event, that data is permanently lost. Because we are forced to deal with these inevitable sensor failures, we must choose how to represent that missingness to the model.
>
> Currently I am missing the discussion about distorted imputations and modal collapse in the Manuscript.
>
> **Change Made:** A paragraph on this has been added to the introduction and discussion,

---

### Review · Reviewer_q1h1 · 2026-04-28

**Summary Of Contributions:**

This paper proposes Min-Std, a simple yet effective imputation strategy for handling missing values in LSTM-based time series forecasting. Instead of reconstructing missing values with model-based estimates (e.g., Kalman Smoothing, MissForest), the authors replace NaNs with an extremal sentinel value that maps exactly to 0 under Min-Max scaling, enabling the LSTM to learn that 0 represents uninformative input. The key contributions are:

1) proposed Min-Std imputation method and its computation cost
2) a large-scale standardized benchmark
3) the release of a new open-source dataset

**Audience:**

Yes

**Audience Explanation:**

Missing values are a ubiquitous problem in real-world time series applications, and the finding that a near-zero-cost sentinel-based method can match or outperform expensive model-based imputation is practically significant. The idea of encoding missing information in representation-space rather than attempting reconstruction is conceptually appealing and could inspire extensions to other architectures and domains.

**Claims And Evidence:**

Yes

**Claims Explanation:**

Extensive experiments provides convincing support that Min-Std performs at least as well as complex model-based alternatives within the tested domain. However, the paper claims that 0 values from Min-Std and 0 values from dropout jointly teach the model to treat 0 as uninformative, yet the described architecture applies dropout between LSTM layers rather than at the input level. The core empirical finding stands, but the theoretical motivation needs revision.

**Requested Changes:**

1) The paper's central motivation that Min-Std zeros and dropout zeros jointly teach the model to treat 0 as uninformative is contradicted by the architecture (Section 3.1.1), which applies dropout between LSTM layers, not at the input. It would be nice to see an ablation study on this.

2) The paper should position against missingness-aware architectures, not only classical imputation methods.

3) Justification for k=1 beyond empirical ranking.

4) Discussion of information loss from collapsing all variable-wise missingness into a single sentinel value (0), versus explicit binary missing masks as additional input channels.

---

> ### Author Response · Authors · 2026-05-22
> **Responses to Reviewer 4 (q1h1)**
>
> **Note:** There is a “Common Themes” comment on the paper that addresses comments that are similar across reviewer comments. It is intended to reduce repetition, and will be referenced within individual responses.
>
>
> # Requested Changes:
>
> 1. The paper's central motivation that Min-Std zeros and dropout zeros jointly teach the model to treat 0 as uninformative is contradicted by the architecture (Section 3.1.1), which applies dropout between LSTM layers, not at the input. It would be nice to see an ablation study on this.
>
> **Response:** See ‘Common Themes’ comment Section ‘4) Dropout’ for response.
>
> 2. The paper should position against missingness-aware architectures, not only classical imputation methods.
>
> **Response:** See ‘Common Themes’ comment Section 2\) Baseline Imputation Strategies for response.
>
> 3. Justification for k=1 beyond empirical ranking.
>
> **Response:** See ‘Common Themes’ comment Section 1\) K-factor for response.
>
> 4. Discussion of information loss from collapsing all variable-wise missingness into a single sentinel value (0), versus explicit binary missing masks as additional input channels.
>
> **Response:** We acknowledge that collapsing missingness into a single representation-space channel theoretically loses the explicit variance separation provided by multi-channel binary masks. We have updated our Conclusion (Section 7\) to explicitly outline future work comparing multi-sentinel single-channel representation-space missingness encoders against multi-channel binary missing-masks and time-since-last-seen features.

---

### Author Response · Authors · 2026-05-22
**Common Themes**

# Common Themes
**Note:** The four points below are to address common themes across reviewer comments. It is intended to reduce repetition, and will be referenced within individual responses.

---

> ### Author Response · Authors · 2026-05-22
> **1) k-factor**
>
> **Reviewer 1 (N8BS):**  About the k (buffer) factor: I think it would be rather easy to check, even only on the train set, how much it is likely to have values exceeding the buffer: Doing cross-validation (split in train/val), one could look at the percentage of  validation examples that are beyond the Min-Std "fitted" on the train set, and average over folds. I guess a good k is one where such events are extremely rare. An extra buffer that helps the network really flag NaNs as "out of domain" may actually also help the LSTM. This would be interesting, as it would help understand the origin of the success of the Min-Std imputation: it is enough to have no test value exceeding the threshold, or it it better that NaNs be far away form actual values ?
>
> **Reviewer 2 (AQCf):** There is no mathematical or experimental work that leads them to make the decision on the mean \- 2 \* std. Why this particular value? Have they systematically tried other values to reach this? I only saw mean \- 1\* std. \[NOTE: the authors believe reviewer 2 (AQCf) meant min \- k \* std, not mean \- k\* std, as our paper does not mention the latter \]
>
> **Reviewer 4 (q1h1):** Justification for k=1 beyond empirical ranking.
>
> **Response:** We have run the analysis (Appendix A table 4 and B table 7\) suggested by reviewer 1 (N8BS) which confirms the k=1.0 choice. We performed this on cross-validation (5 folds in training set) as requested, and also using the original split fitting min and std on 70% train. For 4 values of k {0.25, 0.5, 1, 2} we calculate the count/percent of values that fall below the “fitted” sentinel (buffer exceedances), the worst exceedance percentage, and the percent of transformed space reserved for buffer. We also calculate the value of k required to have no buffer exceedances. This analysis was performed for the Air Quality and GRWS datasets separately.
>
> From this we see that k=0.25 is too small, both GRWS and Beijing Air Quality have values in val/test that would fall below the sentinel (buffer exceedances). k=0.5 eliminates all buffer exceedances in GRWS but not Beijing Air Quality. k=1 eliminates all sentinel exceedances across all datasets with the original validation/test holdout. Note that k=0.5 eliminated nearly all (7 occurrences remain from \~5,000,000 observed values) sentinel exceedances, yet the LSTM prediction results were superior with k=1 compared to k=0.5.
>
> Testing values larger than 1 may be considered for future work, however as we raise this scaling factor, we also increasingly compress the range of real values after scaling and thus lose precision. To address reviewer 2’s (AQCf)  specific mention of k=2 we include k=2 in the newly added tables, however we have not included additional models trained/tested with k=2 at this point as k=1 is large enough to prevent buffer exceedances using the original validation/test holdout.
>
> **Change Made:** Two tables are added to the appendices, a table for GRWS in Appendix A, and for Air Quality datasets in Appendix B. A brief mention of these tables is added to the final paragraph of the discussion section.

---

> ### Author Response · Authors · 2026-05-22
> **2) Baseline Imputation Strategies:**
>
> **Reviewer 2 (AQCf):** There are also many powerful methods that have not been tested or even mentioned such as imputation \+ flagging, modulation layer ([https://openreview.net/forum?id=MRLHN4MSmA](https://openreview.net/forum?id=MRLHN4MSmA)), GRAPE ([https://arxiv.org/abs/2010.16418](https://arxiv.org/abs/2010.16418)), VAEAC ([https://openreview.net/forum?id=SyxtJh0qYm](https://openreview.net/forum?id=SyxtJh0qYm)), Neumiss ([https://arxiv.org/abs/2007.01627](https://arxiv.org/abs/2007.01627)).
>
> **Reviewer 3 (x3TT):** “The Authors should comment on the role of modal collapse in generative filling of missing points.” and “Using the output of trained AI solutions to fill gaps have tendency to increase the probability of certain types of imputations, that may increase the probability of the inputed choices with respect to other entries leading to a distorted predictions of probabilities of the next entries.”
>
> **Reviewer 4 (q1h1):** The paper should position against missingness-aware architectures, not only classical imputation methods.
>
> **Response:** The objective of the submitted paper, *Learning from Missing Values: Encoding Missingness in Representation-Space for LSTM Time Series Forecasting,* is to introduce and validate the viability of using single-channel, explicit Representation-Space Missingness Encoders for imputation on the LSTM model. A core thesis of our work demonstrates that a simple and computationally trivial representation-space encoding can yield superior predictive performance without the substantial overhead of additional generative model training. Therefore, we focus our selection to meet four criteria: (1) single-channel (2) stand-alone preprocessing imputation methods that (3) naturally integrate with the LSTM sequence model and (4) time-series prediction.
>
> Reviewer 2 (AQCf) and 4 (q1h1) helpfully suggest utilizing advanced missingness-aware architectures. However, the architectural intervention methods suggested (GRAPE, NeuMiss, and Modulation Layers) are separate models from the LSTM, and we consider them outside the scope of this study (per criteria 2&3 above). This study specifically isolates and evaluates preprocessing representations and imputation under a controlled, fixed LSTM architecture to ensure an apples-to-apples comparison. The suggested methods are architectural interventions, not preprocessing strategies. Comparing, for example a Bipartite GNN (GRAPE), to an LSTM would confound the variables, making it a challenge to tell if performance gains/losses are due to the handling of missingness, or the change to a different predictive model architecture.
>
> Unlike the above suggestions, VAEAC can function as a classical stand-alone preprocessing imputer. We have included methods (KNN, MissForest) in our study that, like VAEAC, assume tabular i.i.d. data and lack any native temporal mechanisms. We also included Kalman, which is built explicitly for dynamic, time-varying systems. We have investigated VAEAC as a potential imputation baseline; however, unlike the modelling methods we selected, which can be deployed generically across datasets using publicly available frameworks, the standard VAEAC implementation does not have separate fit() and transform() functions allowing the user to specify what exact data are to be used while fitting the model. Instead, the available implementation (not intended for timeseries violating criteria 4\) takes a random selection of samples to use as validation vs train (which destroys chronology). If the available code were modified to respect chronological splits rather than random, we would still have data leakage, as there is no option to freeze the model to impute data on the ‘test’ portion of the data. This means even with significant modification to the suggested method to accommodate time-series, the training will be performed using our test set as validation causing information leakage (global scaling via norm\_mean and norm\_std, test-driven early stopping, etc).
>
> Reviewer 3 (x3TT)  highlights a critical danger related to modal collapse: using model-based or generative AI solutions to reconstruct these gaps can indeed distort the empirical distribution of the training data, especially for variables such as precipitation where the missing value may not be predictable from available covariates. This exact danger is a primary motivation for our Min-Std method. This ensures the downstream network learns from the actual observed data and the structural pattern of missingness.

---

> > ### Author Response · Authors · 2026-05-22
> > **2) Baseline Imputation Strategies (Continued)**
> >
> > Inclusion of multi-channel missing-flag and time-since-last-seen variables violates criteria (1) and was not included in this study. However, future work is to consider Min-Std as a member a new class of “Representation-Space Missingness Encoders” (RSME), this will include causal gap-aware multi-sentinel RSME, where different sentinel values encode whether a missing value is the first missing value in a gap, or part of an ongoing missing run, with optional duration-dependent sentinels based on the time-since-last-seen value. The addition of the above multi-channel approaches will be more appropriate when compared to a future gap-aware RSME.
> >
> > **Change Made:** Future work in conclusion expanded to more clearly place this study, and introduce the future gap-aware Representation-Space Missingness Encoders with comparison to flagging and time-since-last-seen. Introduction and discussion are expanded to discuss the concern of modal collapse with model-based imputation.

---

> > ### Comment · Reviewer_N8BS · 2026-06-10
> > **Literature review**
> >
> > I think the points raised by the other reviewers about relation to pre-existing works were fair.
> > I see you have updated the manuscript, but I wuld suggest to:
> > - cite the papers when referring to the methods (GRAPE, VAEAC) (currently they are mentionned without reference to the papers)
> > - extend a bit the discussion in hte manuscript, by including some of the disussion you have put in your answetrs to us.  The current discussion in the manuscript is short, and comes quite late, when it is actually an important point for future readers, esp. new to the field.
> > Note that it is ok to criticize prior works (for how expensive they are, etc) in your paper.

---

> ### Author Response · Authors · 2026-05-22
> **3) Number of experiments:**
>
> **Reviewer 1 (N8BS):** Could you explicit the number of experiments that are summarized in the box plot of fig 1 and following figures? I read N=112 for the Air quality dataset, nothing for GRWS. N=112 comes from (\#time series)x(\#target values)x(\#time horizons) ? It's important to explicit it, it'd help understand the manuscript altogether.
>
> **Reviewer 2 (AQCf):** Given the authors are testing on two datasets only, I am wondering where the N=112 at the top of page 10 is.
>
> **Response:** There are 16 GRWS labels and 12 Beijing Air Quality, each evaluated at 4 timesteps (t+1, t+2, t+3, t+4), the N=112 comes from (16+12)\*4=112. Note the GRWS has 17 total datasets (thus 17 label timeseries), however one was used exclusively for hyperparameter tuning and not included in results. Note that there are 2 *categories* of datasets, one containing 16 datasets and another containing 12 datasets, so 28 datasets in total.
>
> **Change Made:** Updated text in S5.2 to “GRWS ($N=64$) and Air Quality ($N=48)$ for a total $N=112$”

---

> ### Author Response · Authors · 2026-05-22
> **4) Input Dropout**
>
> **Reviewer 1 (N8BS):**  There is some confusion on the Dropout analogy.
>
> * In the introduction, there is some mention of Dropout acting on features, without the detail that the dropout considered is *input* Dropout. This is confusing, as it leads to think that the paper is confusing inputs fed to the LSTM and activations within the LSTM. Please mention *input* dropout earlier in the manuscript, to avoid this confusion, if this is really what is done.
> * but then, in sec 3.1.1, I do not see any input dropout used, in the architecture, it's actually in between LSTM layers that Dropout is used. This fact challenges the interpretation made in the earlier sections, stating that turning NaNs into 0's (after inputation+transform) is somewhat equivalent to (input) dropout.
>
> So, either:
>
> * correct the architecture in 3.1.1, adding input dropout, and mention the "input" keyword early on, if it's what was done.
> * or, (most likely), correct the motivation in the introduction. The input values (attributes) coming from the data (and that are imputed with Min-Max here) do not have the same intrinsic meaning as features (embeddings) living inside the networks (values of the activations coming out of neurons). Dropout sets activities to 0 because 0, for most activation function, is a natural "no activity", neutral state, that just decreases the sum of activities received by the next neuron. In contrast, input values (dataset attributes, also called features sometimes) have no reference 0\. After Min-Max scaling, 0 is the smallest possible value, but is not "neutral" either.
>
> **Reviewer 4 (q1h1):**  The paper's central motivation that Min-Std zeros and dropout zeros jointly teach the model to treat 0 as uninformative is contradicted by the architecture (Section 3.1.1), which applies dropout between LSTM layers, not at the input. It would be nice to see an ablation study on this.
>
> **Response:** The reviewers have pointed out an important flaw in our original submission.  By discussing input dropout early in the paper we gave the incorrect impression that we included input dropout in the LSTM model used in this study. Upon re-reading the paper, we see how this was not explained clearly. So, in this resubmission, we have revised/removed all text (helpfully identified by Reviewer 1 (N8BS)) that gave this impression. We have ensured the introduction clearly states that we do NOT use input dropout.
>
> The incorrect impression we gave stems from the fact that input dropout partially inspired the concept of min-std in conjunction with minmax scaling. Our objective was an imputation method that was NOT attempting to guess missing values, but rather using a value logical for the model. When framing the idea of encoding missingness into input-space, we considered that input dropout assigns 0 values, and min-std was designed to ensure imputed values are similarly 0\.
>
> For the model used in this paper, many hyperparameters/model architecture were selected using Optuna Bayesian Optimization (number of layers, regularizers, LSTM units, etc.). These optuna trials were performed on a single GRWS dataset (Leggatt) with Zero Imputation. I selected Zero Imputation as a baseline during optuna trials to ensure fairness across imputation methods. Early initial min-std testing DID include input dropout, however the architecture selected by Optuna did not suggest input dropout, and thus the experiments in this paper do not employ input dropout.
>
> Future work with encoding missingness into input space will include ablation study on the impact of including an input layer dropout.
>
> **Change Made:** The sentences regarding dropout noted by reviewer 1 (N8BS) have been changed to show they are inspired/motivated by dropout, and not implying that input dropout is used in this study.